# Non-cooperative 4E-BP2 folding with exchange between eIF4E-binding and binding-incompatible states tunes cap-dependent translation inhibition

Jennifer E. Dawson [1,8], Alaji Bah[1,2,8], Zhenfu Zhang[3], Robert M. Vernon[1], Hong Lin[1], P. Andrew Chong [1], Manasvi Vanama[1], Nahum Sonenberg[4,5], Claudiu C. Gradinaru [3,6] & Julie D. Forman-Kay [1,7 ✉]

Phosphorylation of intrinsically disordered eIF4E binding proteins (4E-BPs) regulates cap-dependent translation by weakening their ability to compete with eIF4G for eIF4E binding within the translation initiation complex. We previously showed that phosphorylation of T37 and T46 in 4E-BP2 induces folding of a four-stranded beta-fold domain, partially sequestering the canonical eIF4E-binding helix. The C-terminal intrinsically disordered region (C-IDR), remaining disordered after phosphorylation, contains the secondary eIF4E-binding site and three other phospho-sites, whose mechanisms in inhibiting binding are not understood. Here we report that the domain is non-cooperatively folded, with exchange between beta strands and helical conformations. C-IDR phosphorylation shifts the conformational equilibrium, controlling access to eIF4E binding sites. The hairpin turns formed by pT37/pT46 are remarkably stable and function as transplantable units for phospho-regulation of stability. These results demonstrate how non-cooperative folding and conformational exchange leads to graded inhibition of 4E-BP2:eIF4E binding, shifting 4E-BP2 into an eIF4E binding-incompatible conformation and regulating translation initiation.

---

[1] Program in Molecular Medicine, The Hospital for Sick Children, Toronto, ON M5G 0A4, Canada. [2] Department of Biochemistry and Molecular Biology, SUNY Upstate Medical University, Syracuse, NY 13210, USA. [3] Department of Chemical and Physical Sciences, University of Toronto, Mississauga, ON L5L 1C6, Canada. [4] Department of Biochemistry, McGill University, Montreal, QC H3G 1Y6, Canada. [5] Goodman Cancer Research Centre, McGill University, Montreal, QC H3A 1A3, Canada. [6] Department of Physics, University of Toronto, Toronto, ON M5S 1A7, Canada. [7] Department of Biochemistry, University of Toronto, Toronto, ON M5S 1A8, Canada. [8] These authors contributed equally: Jennifer E. Dawson, Alaji Bah. ✉email: forman@sickkids.ca

Tuning the binding affinities of the eukaryotic translation initiation factor 4E (eIF4E) for its intrinsically disordered binding proteins (4E-BPs) is central to regulation of cap-dependent translation initiation. Initiation is the rate-limiting step in translation, in which the ribosome is recruited to the mRNA by the eIF4F complex[1–3]. eIF4E, along with the RNA helicase eIF4A and the scaffolding protein eIF4G, forms the eIF4F complex. eIF4E directly interacts with the 7-methyl guanosine cap structure at the mRNA 5′ end. Interaction of eIF4E with 4E-BPs[4,5] inhibits cap-dependent translation initiation by competing for an overlapping eIF4E surface with eIF4G[6–9]. 4E-BPs have multiple phospho-sites[10] (four or more depending on isoform or species), which are hierarchically phosphorylated[3,11]. Non-phospho 4E-BPs (np-4E-BPs) bind eIF4E tightly and inhibit eIF4G binding[10,11]. The mammalian target of rapamycin (mTOR) phosphorylates the first two sites, T37 and T46[12], resulting in the hypo-phosphorylated state, which binds eIF4E more weakly, but still inhibits eIF4G binding. Hyper-phosphorylated 4E-BP is modified at all sites, including S65 and T70, and has further weakened eIF4E affinity, allowing eIF4G to compete for eIF4E and translation initiation to proceed[10]. The hyper-phosphorylated state has a potential fifth site at S83 that is conserved in the 4E-BP1 and 4E-BP2 isoforms, but is less conserved in 4E-BP3 and in invertebrates[13]. Since dysregulation of eIF4E function is involved in many diseases including cancer and autism spectrum disorders[1,2] and ubiquitin-mediated degradation of 4E-BPs is dependent on its phosphorylation status[14], understanding the stepped binding affinities and the link with protein stability is critical.

4E-BPs, including the predominantly neuronal isoform 4E-BP4,5, are intrinsically disordered proteins (IDPs)[15–17], yet contain significant transient secondary structure distributed throughout the protein[8,11]. The canonical binding helix is conserved between eIF4G and the 4E-BPs (⁵⁴YDRKFLLDRR⁶³ for 4E-BP2) and binds to the same interface on the convex surface of eIF4E structures of the 4E-BP:eIF4E complexes[7,9,13,18–20]. The residues after the helix are less conserved and more dynamic, and contain a linker region and secondary binding site (centered at ⁷⁸IPGTV⁸² in 4E-BP2), which winds along the lateral surface of eIF4E[7–9,13]. Determining structural effects of phosphorylation is complicated by available structures of 4E-BP:eIF4E complexes being for np-4E-BP fragments with few phospho-sites present in the observed structured regions, and only one site, S65, being in direct contact with the eIF4E surface. Proximity of S65 to E70 on eIF4E suggested the potential for electrostatic repulsion between the binding partners[18,21], but the S65 sidechain is oriented differently relative to the eIF4E surface in different crystal structures, pointing to binding being controlled by other mechanisms[11,22].

Although a low-resolution model based on SAXS data exists[23], no atomic structure of eIF4E in complex with any full-length 4E-BP has been observed, despite intense structural studies since 1998[17]. Interestingly, only upon truncating both the N-terminal and C-terminal disordered regions (residues 44–87[21] and residues 50–83[13]) or abrogating the secondary binding site in the context of full-length protein[8] (⁷⁸IPGTV⁸² deleted or mutated to ⁷⁸AAAAA⁸²), did NMR spectroscopy studies and X-ray crystallographic structures begin to reveal detailed structural and dynamic interactions between eIF4E and the extended bipartite 4E-BP-binding site[17]. Together, these data demonstrated that full-length 4E-BP:eIF4E interactions are dynamic, fuzzy complexes[17].

We have previously demonstrated that the phosphorylation of T37 and T46 in apo 4E-BP2 induces folding of residues P18–R62 into a four β-stranded folded domain that partially sequesters the canonical eIF4E-binding helix, with pT37 and pT46 being part of conserved pTPGGT motifs that form hairpin turns central to the structure[11]. The C-terminal intrinsically disordered region (C-IDR, residues 63–120) contains the secondary eIF4E-binding site, the linker region between the canonical and secondary binding sites, and three additional phospho-sites (S65, T70, and S83). The C-IDR is not required for the folded domain formation[11]. However, phosphorylation at T37 and T46 alone only reduces 4E-BP2's eIF4E affinity by ~100-fold compared to np-4E-BP2 (from $K_D = 3.2 ± 0.6$ to $267 ± 32$ nM). 4E-BP2 phosphorylated at all five sites has ~4000-fold weaker affinity ($K_D = 12,320 ± 200$ nM), allowing eIF4G to out-compete 4E-BP2, indicating that the three C-IDR phospho-sites also play important roles in binding[11]. We previously found that five-phospho 4E-BP2 (5p-4E-BP2) mutants that lacked folded domains have affinities similar to that of np-4E-BP2, establishing that the C-IDR sites must act in concert with the folded domain[11]. This, together with evidence that the folded domain is marginally stable and is in exchange with partially unfolded states[11], led to the hypothesis that C-IDR phosphorylation stabilizes the folded domain and that increased stability lowers 4E-BP2:eIF4E binding by decreasing availability of the eIF4E-binding helix.

Here, we define in mechanistic detail how the C-IDR phospho-sites modulate the folded domain stability, with hierarchical phosphorylation controlling eIF4E binding by tuning access to eIF4E-binding sites via changes in conformational equilibria. Using NMR spectroscopy, single-molecule fluorescence and calorimetry, we show that stability of the two pTPGGT hairpin turns engenders non-cooperative folding of the phosphorylated domain that controls access to the canonical binding site, while C-IDR phospho-sites tune the domain's stability. The biophysical stability of the pTPGGT turns, combined with biochemical and bioinformatic results, suggests a more general relationship of pTPGGT-like motifs to cellular degradation and stability. Overall, our data explain the underlying mechanism of graded inhibition of eIF4E binding by hierarchical phosphorylation of 4E-BP2, which regulates the protein's dynamic conformational equilibria, gradually converting it into an eIF4E binding-incompatible conformation.

## Results

**Binding largely unaffected by phosphate electrostatics**. Previously, we found that the presence of the folded domain was indispensable for weakening eIF4E:4E-BP2 binding. Mutants that disrupt the folded domain of 5p-4E-BP2 bind eIF4E with similar affinities to that of np-4E-BP2, arguing against a dominant effect of overall electrostatics[11]. However, structures of eIF4E bound to closely related 4E-BP1 fragments show the close proximity of phospho-site S65 to a negative charge on eIF4E, suggesting that phosphorylation of S65 may weaken binding by electrostatic repulsion. Analysis of the energy minimized bound-state structure (PDB ID 4UED), with and without phosphate, revealed no detectable energetic difference (Supplementary Fig. 1b, c). Thus, modeling coulombic and steric interactions due to phosphorylation at S65 in bound state structures does not capture the effect of phosphorylation.

**Phospho 4E-BP2 folded domain unfolds non-cooperatively.** Understanding binding energetics requires consideration of the equilibrium between all possible states, not only analysis of the bound state. We tested our primary hypothesis that C-IDR phosphorylation affects binding by modulating the 4E-BP2 folded domain stability. We used NMR spectroscopy and differential scanning calorimetry (DSC) to measure the stability of the fold under different denaturing conditions and phosphorylation states. Despite the folded domain's small size, ~40 residues, we found that it is not a simple two-state folder. The presence of non-cooperative folding was readily observed in NMR ¹⁵N-¹H

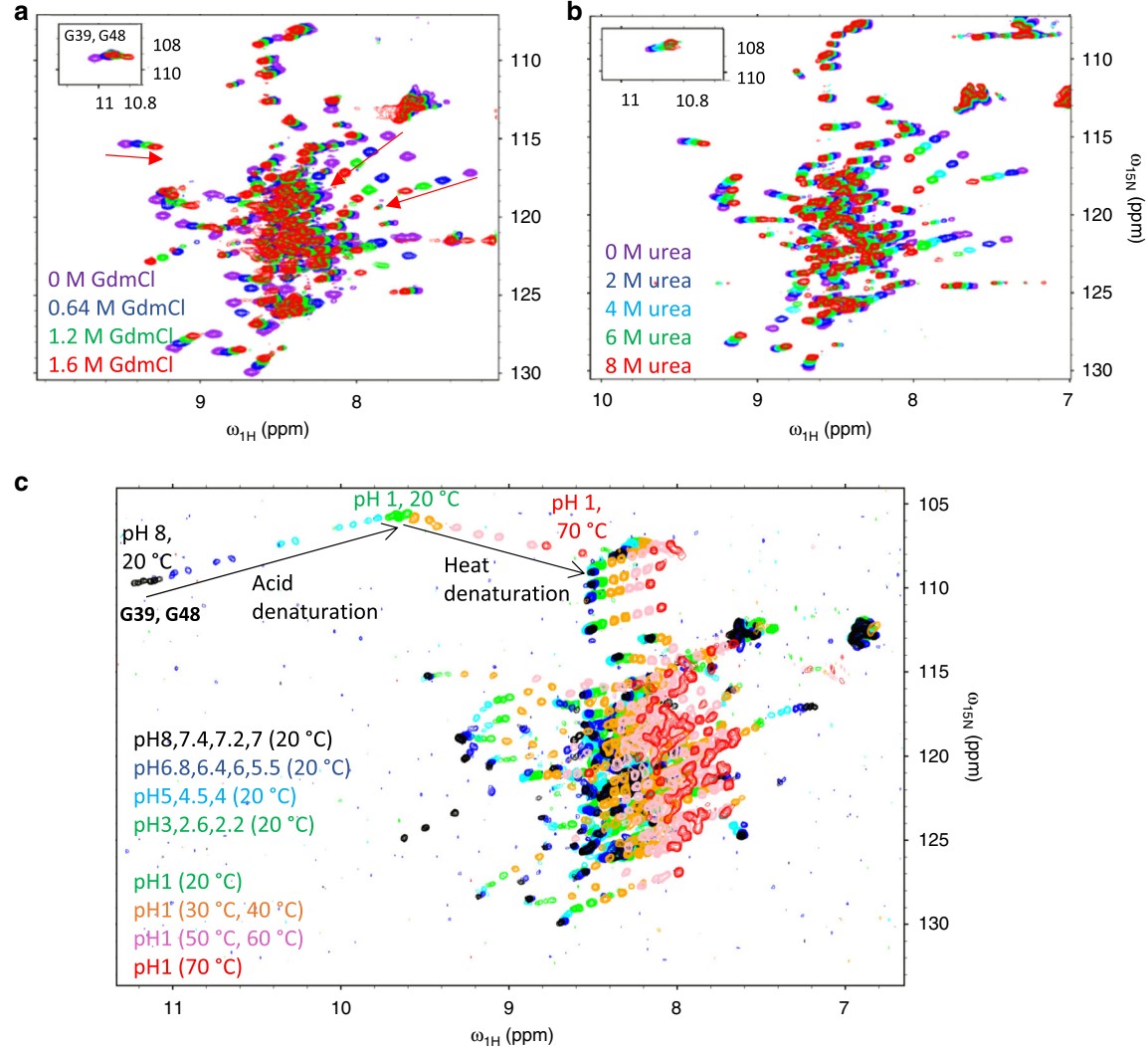

**Fig. 1 Chemical, acid, and thermal denaturation of five-phospho 4E-BP2. a** Overlay of NMR $^{15}$N–$^{1}$H HSQC spectra of 5p-4E-BP2 GdmCl titration samples at pH 6.0 and 20 °C. **b** Overlay of NMR $^{15}$N–$^{1}$H HSQC spectra of 5p-4E-BP2 urea titration samples at pH 6.0 and 20 °C. Insets to **a** and **b** panels contain the downfield portion of the spectrum with resonances of G39 and G48. **c** Acid and thermal denaturation of 5p-4E-BP2. The acid denaturation data shown were collected at 20 °C between pH 8.0 and pH 1.0 (spectra colored in gradient from black to green). Thermal denaturation at pH 1.0 is shown for temperatures between 20 and 70 °C (gradient from green to red). The two arrows in the upper left corner of the spectrum show the trajectories of the G39 and G48 resonances for acid and heat denaturation. The peaks on the upper right of the urea spectra **b** are aliased 4E-BP2 amide sidechain resonances, explaining their absence in **a** and **c**.

HSQC experiments on 5p-4E-BP2 (Fig. 1). Matching previous observations[11], chemical shifts for residues in the folded domain fall within ranges associated with secondary structure, clear evidence that the domain is not in a random coil state. The peaks from the hairpin residues G39 and G48 are shifted significantly downfield, reflecting stable backbone NH hydrogen bonds to the phosphate groups. After adding chemical denaturants, guanidinium chloride (GdmCl) or urea (Fig. 1a, b), we observed heterogeneous behavior with most folded domain peaks moving significantly toward random coil values ($\omega_{1H} \sim 8.2$ ppm). In contrast, the hairpin G39 and G48 peaks (Fig. 1a, b insets) remain much farther from random coil values, indicating that the phosphorylation-induced turns are not unfolding with the rest of the domain.

Since deviations from random coil values are still observed for many domain residues even at high denaturant concentration (1.6 M GdmCl or ~8 M urea), we used acid and heat denaturation, yielding peaks much closer to random coil values under acidic (pH 1.0) and high temperature (70 °C) conditions (Fig. 1c).

We also observed non-cooperative unfolding 4E-BP2 by acid and thermal denaturation, with unfolding of the hairpin turns lagging behind the changes in the rest of the domain (Fig. 1c). The chemical shift behavior of the peaks was used to distinguish between residues affected by acid denaturation of the folded domain and the direct effect of temperature and buffer conditions on solvent-exposed residues. Residues that shift toward random coil are likely to be affected by denaturation (Fig. 1c, Supplementary Fig. 2a). In contrast, the peaks for the np-4E-BP2, which does not contain any folded domain, show a uniform shift upfield with increasing temperature, consistent with expectation for non-hydrogen-bonded amides and no conformational changes[24–26] (Supplementary Fig. 2b). To quantify this non-cooperative unfolding for 5p-4E-BP2, we fitted the chemical shift changes at 20 °C as a function of pH to obtain residue-specific apparent p$K_a$ values for the unfolding transition (Supplementary Table 1). Chemical shift changes for residues affected by both the denaturation and the protonation of neighboring solvent-exposed residues were fitted to the three-state equation (see

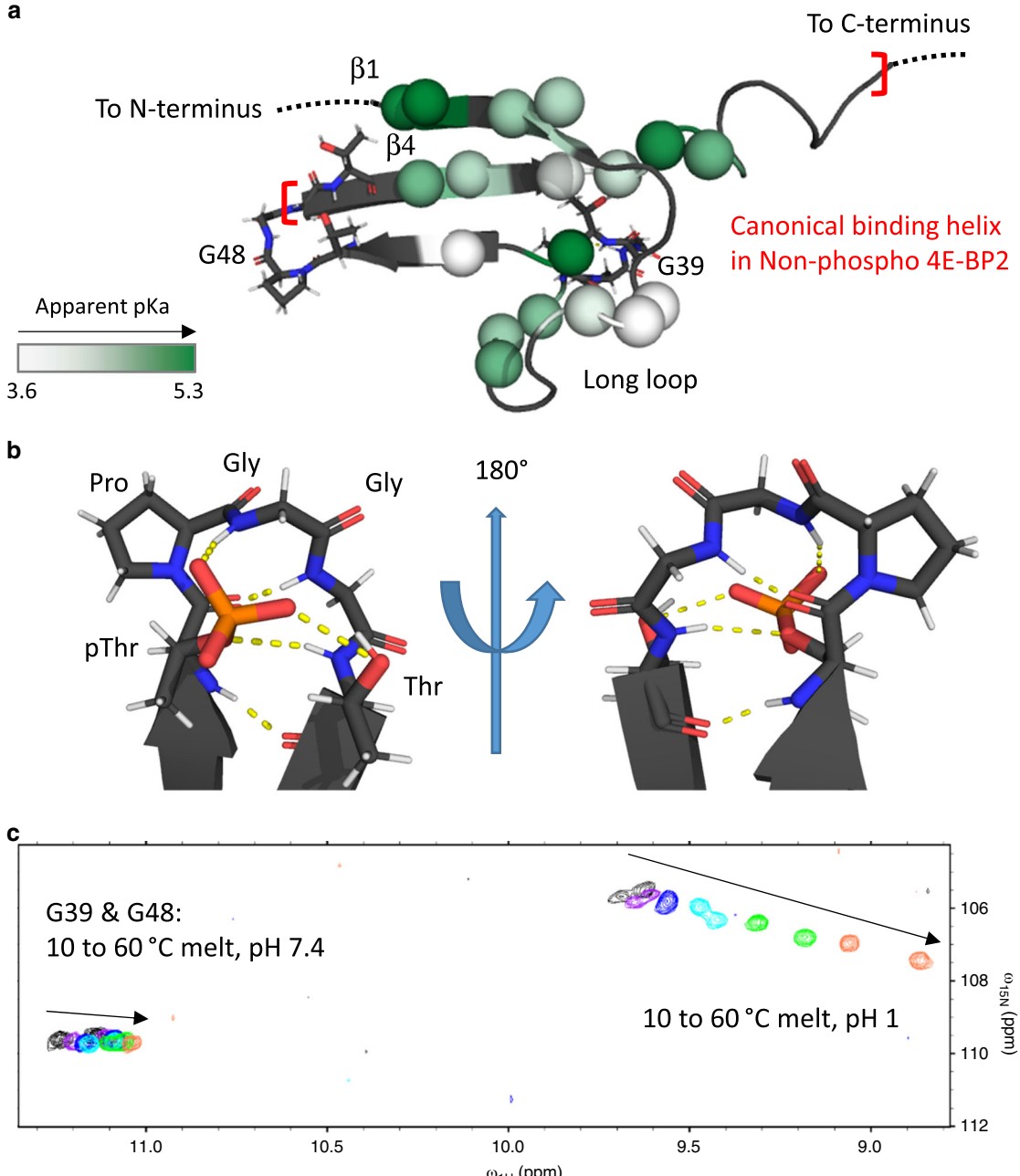

**Fig. 2 Non-cooperative folding of five-phospho 4E-BP2 folded domain. a** Apparent $pK_a$ values from acid denaturation at 20 °C mapped onto residues 18–62 of the 5p-4E-BP2 using a white to green gradient, not including the hairpin turns shown in stick diagrams. The location of the canonical eIF4E-binding helix found in np-4E-BP2 is indicated by red brackets, highlighting that the folded domain disrupts the binding conformation. The variation in apparent $pK_a$ during acid denaturation is consistent with non-cooperative folding of the domain. **b** Hairpin turn for a pTPPGT motif with hydrogen bonds (PDB code 2MX4). **c** Acid and thermal denaturation of hairpin glycine residues, G39 and G48. A selected section of overlaid NMR HSQC data is shown for the thermal denaturation (between 10 and 60 °C) of 5p-4E-BP2 at pH 7.4 and pH 1.0. The hairpin residues have a different unfolding mechanism than the rest of the domain. Source data for Fig. 2a and b are provided as a Source Data file.

"Methods" for further details). If acid denaturation occurred in a two-state cooperative event, a single apparent $pK_a$ for the transition would be found across the domain[27]. Instead, there is a variation in apparent $pK_a$ values between 3.62 and 5.40 (Fig. 2a, Supplementary Table 1), with higher apparent $pK_a$ values indicative of less structural stability. This non-cooperative behavior extends beyond the hairpins, with differences in stability distributed across the domain.

NMR data could only be collected up to 70 °C or 1.6 M GdmCl due to NMR probe engineering limits, so we used DSC to measure

overall melting transition ($T_m$) values of 84 °C at pH 7.4 and 49 °C at pH 1.0 (Supplementary Fig. 3a). We also developed a method to indirectly estimate the thermal denaturation free energies from NMR thermal melting data (between 5 and 70 °C), collected at six different pH values between pH 2.6 and 7.4. Our method, based on the temperature-dependence of the $pK_a$ values (see "Methods" section, Supplementary Fig. 3b–e), reveals that residues near the hairpin hydrogen bonds between the pT37 phosphate and G39 NH and the pT46 phosphate and G48 NH display pH-dependent thermal stability (Fig. 2b, Supplementary Fig. 3e). Other residues in

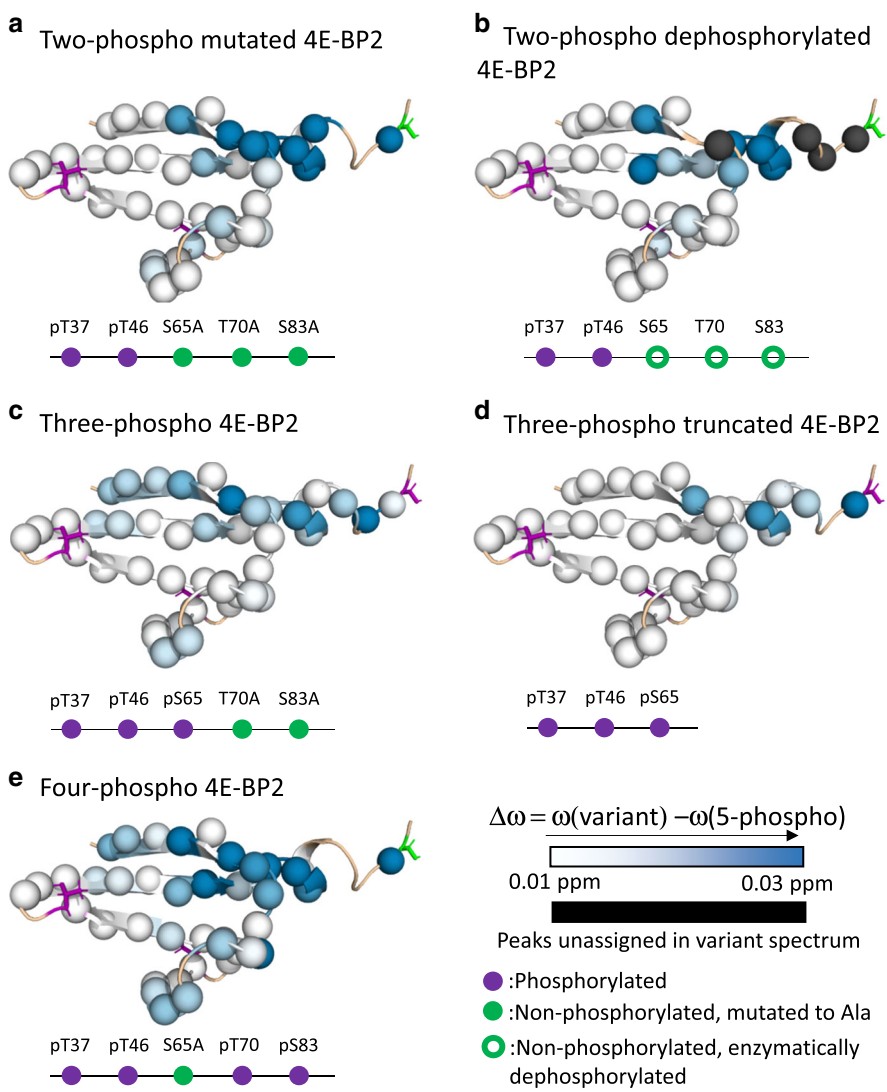

**Fig. 3 Effects of the C-IDR and its phosphorylation status on the 4E-BP2 folded domain.** Changes in backbone amide chemical shifts, $\Delta\omega$ (here referenced to 5p-4E-BP2) indicate that the structure and dynamics of 5p-4E-BP2 were perturbed by modifications, either by mutations that preclude full phosphorylation or de-phosphorylation. The $\Delta\omega$ values are mapped onto residues 18–66 of 4E-BP2 structure as a white-to-blue gradient. The analyzed residues are shown as spheres. The positions of the phosphates (purple stick diagrams, purples circles on the schematic) and alanine mutations that cannot be phosphorylated (green stick diagrams, circles on the schematic) are indicated. For further details, see Supplementary Fig. 4. **a** 2p-4E-BP2 phosphorylated at T37 and T46, with S65A/T70A/S83A mutations. **b** Wildtype 2p-4E-BP2. 5p- (unmutated) 4E-BP2 was de-phosphorylated to 2p-4E-BP2 using lambda protein phosphatase. Residues with missing peaks in the four-hour de-phosphorylation $^{15}$N-$^1$H HSQC are shown as black spheres (residues 25, 62, 63, and 65). **c** 3p-4E-BP2 phosphorylated at T37, T46, and S65, with T70A/S83A mutations. **d** 3p-truncated 4E-BP2 containing residues 1–67 of 4E-BP2, phosphorylated at T37, T46, and S65. **e** 4p-4E-BP2 (full-length) 4E-BP2 phosphorylated at T37, T46, T70, and S83, with a S65A mutation. Source data are provided as a Source Data file.

the folded domain do not show this behavior, demonstrating the hairpins' independence as miniature folding units. The pH dependence of hairpin stability (Fig. 2c) suggests that protonation of the phosphate weakens the critical hydrogen bond to the first glycine of the hairpin motif (see "Methods" section). We reported earlier that the folding of the hairpins upon T37 and T46 phosphorylation was a prerequisite for folded domain formation[11]. But this was not sufficient for its stability, and the hairpins themselves can be populated even when the domain core is disrupted by mutations[11]. The new data show that the two independently folded hairpins are extremely stable, requiring very low pH to denature, acting as stabilizing molecular staples.

**C-IDR phosphorylation perturbs eIF4E-binding states.** To probe how C-IDR phosphorylation modulates the stability of the folded domain, we systematically replaced the C-IDR sites with alanine substitutions (combinations of S65A, T70A, and S83A). By comparing the NMR spectra of these mutants and 5p-4E-BP2, we observe backbone NH chemical shift differences, $\Delta\omega$, near the C-IDR phospho-sites, but we also observed differences in the long loop connecting strands 1 and 2 of the folded domain, and in the canonical and secondary eIF4E-binding motifs (Fig. 3, Supplementary Fig. 4a, b). Lambda protein phosphatase dephosphorylated S65, T70, and S83 of 5p-4E-BP2 (Fig. 3b, Supplementary Fig. 4c, d), confirming that chemical shift differences are due to phosphorylation status and not due to unintended consequences of the Ala mutations. Removing the S65 site caused the largest chemical shift perturbation of all the modifications, particularly near the canonical binding site residues (Fig. 3). To examine the conformational changes caused by

phosphorylation in more detail, we calculated secondary structural propensity (SSP) values[28] derived from backbone and sidechain chemical shifts ($^1$HN, $^{15}$N, $^1$Hα, $^{13}$CO, $^{13}$Cα and $^{13}$Cβ). This metric estimates the fractional α-helical (SSP > 0) or β-strand (SSP < 0) population in a dynamic state and is used to

gauge the effect of phosphorylation on conformational equilibria. Apo np-4E-BP2 is an intrinsically disordered protein that transiently samples several α-helices along its length, most notably between residues 49 and 67 (Fig. 4a, black bars). These residues encompass the canonical binding motif, partially pre-ordering a

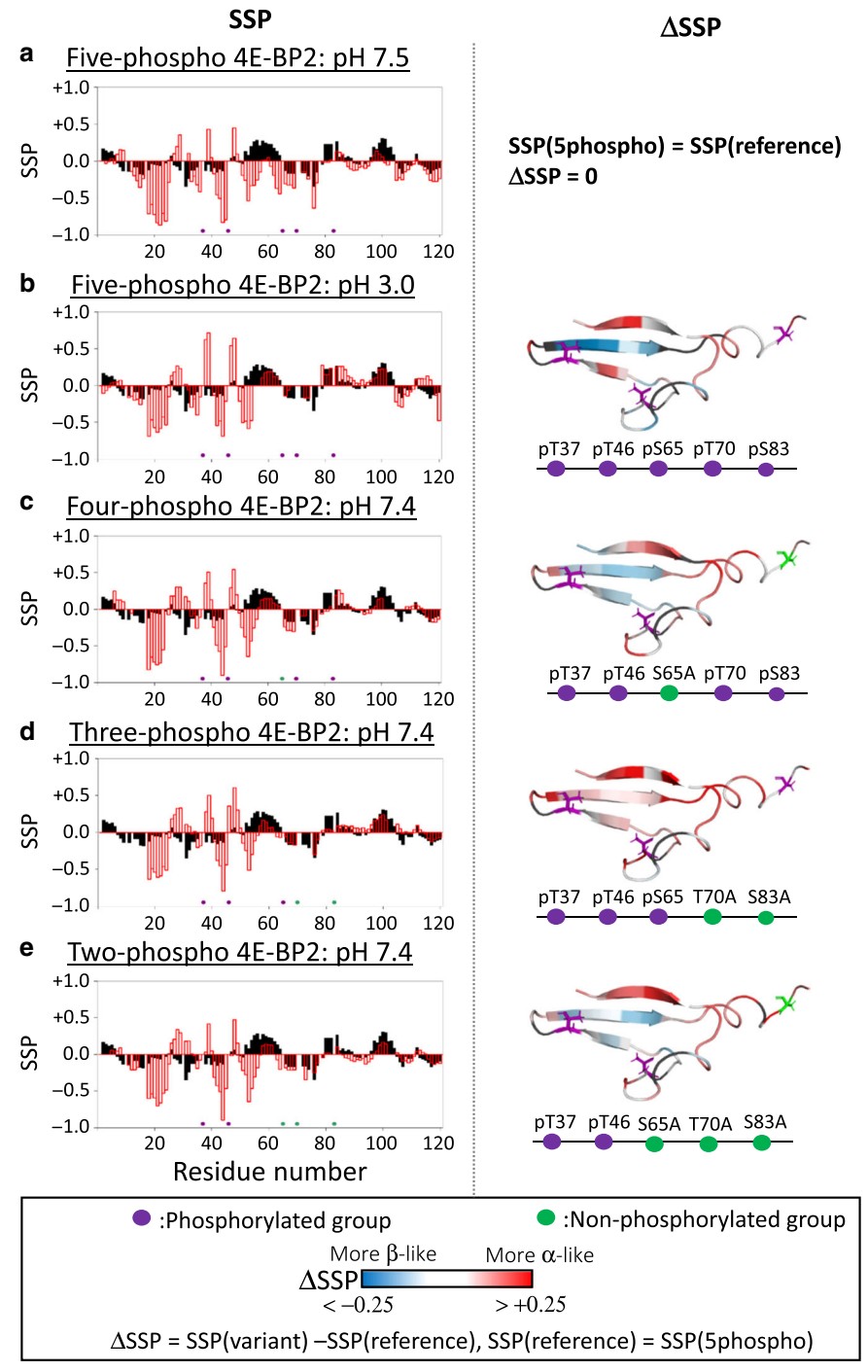

**Fig. 4 Effects of the C-IDR phosphorylation on 4E-BP2 folded domain secondary structure.** The secondary structural propensities (SSP) are sensitive to secondary structural elements and were derived from backbone and sidechain NMR chemical shifts. In the left column, SSP values are plotted versus residue number, where positive values indicate α-helical propensity and negative values indicate β-strand character. In each case, the black bars correspond to np-4E-BP2 SSP at pH 6.0[8]. The red bars are the SSP values for the phospho 4E-BP2 constructs being compared. The locations of the five phospho-sites and the alanine mutations are shown as purple and green stick diagrams, respectively, on the structures and as colored circles on schematics. In the right column, the change in secondary structural propensities (ΔSSP) values due to differences in pH or phosphorylation state were mapped onto the 4E-BP2 structure with increased α-helical propensity in red and increased β-strand character in blue. In each case, the changes are in reference to pH 7.5 5p-4E-BP2 values (making the ΔSSP values for 5p-4E-BP2 at pH 7.5 zero). **a** pH 7.5 5p-4E-BP2 SSP values. **b** pH 3.0 5p-4E-BP2. **c** 2p-4E-BP2 at pH 7.4. **d** 3p- (pT37/pT46/pS65) 4E-BP2 at pH 7.4 (as in Fig. 3c). **e** 4p- (pT37/pT46/pT70/pS83) 4E-BP2 at pH 7.4 (as in Fig. 3e). Source data are provided as a Source Data file.

region that becomes a stable α-helix when bound to eIF4E[8,9]. Np-4E-BP1 in complex with eIF4E also has a helical half-turn elbow loop between residues 64 and 67[13]. Based on our NMR SSP data, much of this binding-ready conformation is abolished in 5p-4E-BP2 (Fig. 4a, red bars). The folded domain β-strands can be observed as four regions with the most strongly negative SSP values between residues P18 and R62, consistent with their being part of the stable folded domain structure. The formation of the folded domain incorporates residues 52–57 of the helical canonical binding motif into its β-sheet structure[11], while the elbow loop region switches from favoring helical to extended or β-strand-like character (i.e., from moderately positive SSP in non-phospho to near zero or negative SSP in 5p-4E-BP2).

The folded domain of 5p-4E-BP2 is most stable at neutral pH with its four β-strands featuring prominently in the SSP data and the rest of the protein tending toward β-strand character. Due to the non-cooperativity of 4E-BP2 folding, parts of the folded domain structure can be altered without undermining the entire domain. Under acid denaturing conditions at pH 3, the β-strand propensities of residues within the folded domain decrease (SSP → 0), especially near the first β-strand, β1, indicating that the fold is destabilized but still partially populated at low pH (Fig. 4b). Some α-helical character has emerged for residues immediately C-terminal to β4, re-forming part of the canonical binding helix or elbow loop, consistent with denaturation yielding a more eIF4E-binding ready conformation for the canonical binding motif. Thus, acid denaturation of phospho 4E-BP2 leads to conformational states closer to those populated by np-4E-BP2 than 5p-4E-BP2.

Partial C-IDR phosphorylation is associated with a range of conformational effects in the folded domain, with helical character adjacent to the canonical binding site decreasing as C-IDR phosphorylation increases (Fig. 4c–e). Matching the chemical shift perturbations, we find that the C-IDR phospho-site at S65 has the largest effect on conformation. Transient helices within the C-IDR are also partially present for the two-phospho, three-phospho, and four-phospho (2p-, 3p-, and 4p-) constructs, indicating that the secondary eIF4E-binding site is also affected. These findings demonstrate a dynamic continuum of conformational states with transient population of eIF4E-binding helices in np-4E-BP2, a conformational equilibrium between these helices and the folded β-sheet structure in partially phosphorylated states and more stable β-sheet structure in the full five-phospho state. Thus, the non-cooperatively folded nature of the folded domain of 4E-BP2 enables its conformational ensemble to be dramatically affected by C-IDR phosphorylation state, controlling the population of eIF4E-binding competent states.

**Phosphorylation affects C-IDR:folded domain interactions**. To further understand how C-IDR phosphorylation could affect the folded domain stability, we monitored transient interactions between the folded domain and the C-IDR using single-molecule fluorescence resonance energy transfer (smFRET) experiments (Fig. 5a) on np-, 2p-, and 5p-4E-BP2. The proteins were labeled with a donor–acceptor dye pair in the long loop of the folded domain (residue 32) and in the C-IDR (residue 91). The average FRET efficiency, $\langle E \rangle$, is inversely proportional to the sixth power of the donor–acceptor distances[29]. In parallel, we recorded NMR paramagnetic relaxation enhancement (PRE) data on 2p-4E-BP2 and 5p-4E-BP2, with a paramagnetic nitroxide label at residue 32 that leads to decreased peak intensity for residues close in space due to resonance broadening[30] (Fig. 5b). Both smFRET and NMR PRE data clearly demonstrate transient contacts that could mediate stabilizing effects of the C-IDR on the folded domain.

smFRET data for np-4E-BP2, 2p-4E-BP2, and 5p-4E-BP2 all show a broad distribution of $\langle E \rangle$ values that shift towards lower values during GdmCl denaturation, indicating an ensemble of conformations expanding and undergoing non-cooperative unfolding[31] (Supplementary Fig. 5). This is in contrast with the decrease in a high-FRET (folded) state accompanied by the increase in a low-FRET (unfolded) state typically observed for cooperative protein unfolding[32]. Together, these data are consistent with our NMR results demonstrating the non-cooperative unfolding of 4E-BP2 for both phospho states. A similar dependency, i.e. gradual shift of $\langle E \rangle$ to lower values (larger distances), is also observed for np-4E-BP2, indicative of unfolding of transient helical structure within the disordered np-4E-BP2 ensemble[8].

In the absence of denaturant at pH 7.4, the FRET $\langle E \rangle$ shifted from 0.57 ± 0.01 for np-4E-BP2, to 0.44 ± 0.01 for 2p-4E-BP2, and to 0.30 ± 0.01 for 5p-4E-BP2 (Fig. 5a, center row). These data indicate that the average distance between residue 32 in the folded domain region and residue 91 in the C-IDR increases with phosphorylation. The mechanism behind this expansion is different than that observed during chemical denaturation. The change in average distance is consistent with the conformations observed previously with NMR. The helical conformations sampled in np-4E-BP2 bring dyes at residues 32 and 91 into close average proximities. While phosphorylation induces folded domain formation, it also causes expansion of the C-IDR. Phosphorylation to 2p-4E-BP2, then to 5p-4E-BP2, shifts the conformational equilibrium towards more extended, β-strand-like conformations (Fig. 4), and leads to an increased separation between the folded domain and the C-IDR. In agreement, NMR PRE data show that residue 32 within the folded domain region and the stretch of residues between 65 and 90 of the C-IDR are farther apart from each other in 5p-4E-BP2 than in 2p-4E-BP2 (Fig. 5b). 5p-4E-BP2 has greater $\langle E \rangle$ at pH 3 than at pH 7.4 (Fig. 5a, right column), with a value, 0.74 ± 0.01, that is similar for all the phospho-states at acidic pH (Fig. 5a, bottom row). This observation is consistent with (i) destabilization (by phosphate group protonation) of all the phosphorylation-induced extended β strands in the folded β-structure and in the C-IDR, and (ii) the subsequent enhanced sampling of helical conformations as detected by NMR spectroscopy[8] (Fig. 4). Together, these two effects enable closer approach of the dyes at low pH for all the different phospho states. Interestingly, for each phospho state, $\langle E \rangle$ values for the dominant peak at pH 10 and pH 7.4 are similar (Fig. 5a, first vs. second row), reflecting little or no change in the protonation status of the phosphates (or other residues). Consequently, there are no significant structural changes, which is consistent with previous observations[11]. Furthermore, we used fluorescence correlation spectroscopy (FCS) to measure the hydrodynamic radius ($R_H$) of 4E-BP2 under different phosphorylation states and pH conditions (Supplementary Fig. 6). At pH 3, $R_H = 27.6 ± 0.6$ Å, independent of phosphorylation state, which is slightly smaller than $R_H$ measured at pH 7.4 ($R_H = 29.0 ± 0.2$ Å) for np-4E-BP2 and $R_H = 31.1 ± 0.2$ Å for 5p-4E-BP2. The FCS data rule out protein aggregation as a cause for high FRET at acidic pH[33]. These data also confirm the observed higher smFRET $\langle E \rangle$ values (i.e. the close distance between residues 32 and 91) for all phospho states of 4E-BP2 at low pH, reflecting the increase in helical population observed in Fig. 4. The longer $R_H$ value for 5p-4E-BP2 at pH 7.4 also agrees with the lower smFRET $\langle E \rangle$ and NMR data showing that folding and stabilization by increased phosphorylation at neutral and high pH reflect extended conformations.

Together, the smFRET, FCS, and NMR PRE results are consistent with the NMR chemical shift-derived SSP data (Fig. 4). A more compact helical structure is observed in np-4E-BP2[8] and

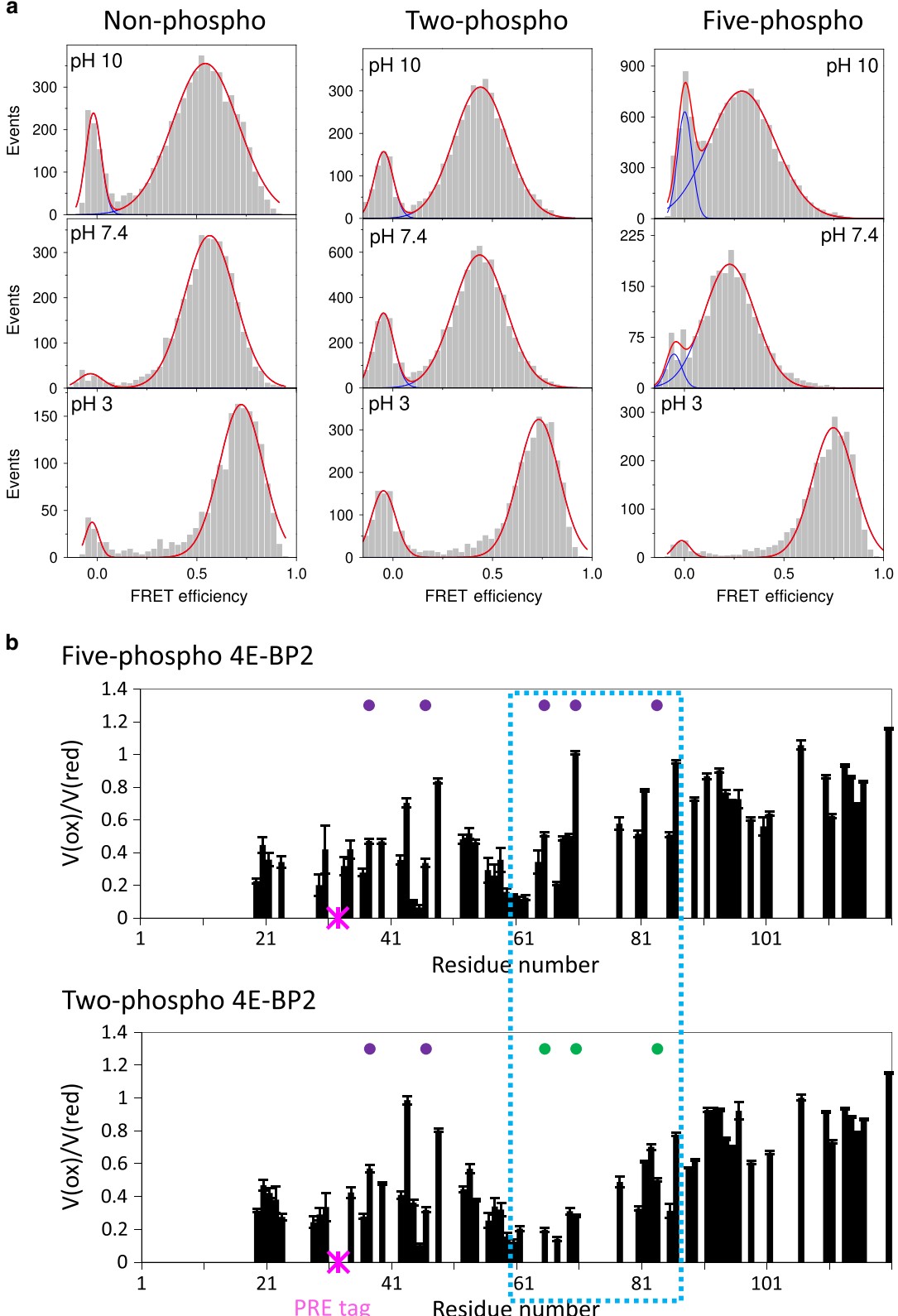

**Fig. 5 C-IDR phosphorylation and pH affect interactions between folded domain and C-IDR. a** smFRET data for acid denaturation of np-4E-BP2, 2p-4E-BP2, and 5p4E-BP2, respectively, with dyes at residues 32 and 91. The FRET efficiency is inversely proportional to the sixth power to the separation between the dyes. The red dashed lines are Gaussian fits to the data, whose maxima indicate the locations of the peak FRET efficiency for the non-zero Gaussian. The average FRET values have an error of ±0.01. **b** NMR paramagnetic resonance enhancement (PRE) data for 2p-4E-BP2 and 5p-4E-BP2 proteins with nitroxide spin labels at residue 32. The attenuation of NMR peak intensities ($V$(ox)/$V$(red)) is proportional to the distance between the nitroxide label and backbone amide groups. The error bars are based on the propagation of the fit heights of the peak volumes. Purple circles indicate phosphorylated residues and green circles are sites of alanine mutations. Source data are provided as a Source Data file.

in phosphorylated 4E-BP2 states under acidic conditions. A more extended β-character, with stabilization of the fold, is found at neutral pH or after C-IDR phosphorylation. Therefore, phosphorylation of C-IDR modulates the structural ensemble of the entire protein, affecting the folded domain stability, transient helical structure within both the folded domain region and the C-IDR, and transient contacts between the folded domain region and C-IDR.

**C-IDR phosphorylation weakens binding via conformational change.** To understand how conformational effects of phosphorylation affect eIF4E-binding affinities, we used isothermal titration calorimetry (ITC) (Fig. 6a, Supplementary Figs. 7 and 8). Our previous work[11] showed that 4E-BP2 mutants with disrupted folded domains bound eIF4E with $K_D$ values of ~2–36 nM, close to the affinity of np-4E-BP2 ($K_D = 3.2 \pm 0.6$ nM)[11] (Supplementary Fig. 7), relatively independent of C-IDR phosphorylation. T37/T46 phosphorylation in 2p-4E-BP2 induces the folded domain and attenuates eIF4E binding to $267 \pm 32$ nM. Phosphorylating the three C-IDR sites in the presence of the folded domain weakens eIF4E binding to $K_D = 15,000 \pm 4000$ nM (ref. [11] and new measurements), demonstrating that the folded domain must be present for phosphorylation of the C-IDR to regulate eIF4E binding. With our new data, we obtained $K_D$ values for eIF4E binding to 4E-BP2 containing the folded domain plus one of the eight possible combinations of C-IDR phosphorylation. 2p-4E-BP2 with no C-IDR phosphorylation and 5p-4E-BP2 with all three C-IDR sites phosphorylated were investigated previously[11]. The other variants (pT37/pT46/pS83, pT37/pT46/pT70, pT37/pT46/pS65, pT37/pT46/pT70/pS83, pT37/pT46/pS65/pS83, pT37/pT46/pS65/pT70) eliminate either one or two C-IDR sites with Ala mutations. All but one of these six 4E-BP2 variants bind eIF4E with $K_D$ values intermediate between that for 2p-4E-BP2 and 5p-4E-BP2 ($K_D \sim$ 1800–5700 nM, Supplementary Fig. 7). The pT37/pT46/pS65/pS83 variant binds eIF4E with an affinity that is statistically similar to that of 5p-4E-BP2. Note that this state is not present in the cell due to hierarchical phosphorylation requiring phosphorylation of T70 before that on S65; it is possible that S65 phosphorylation, with the most dramatic impact on conformational equilibria, requires destabilization from T70 phosphorylation for accessibility of the kinase.

Comparison of the binding affinities with the conformational properties from NMR chemical shifts and SSP calculations provides strong evidence that the effects of phosphorylation on structure and binding are related, consistent with a model in which stabilizing the folded domain disrupts binding and the non-cooperative folding provides a mechanism for titratable stabilization of the folded domain. This structural link between the degree of phosphorylation and binding affinity leads to a tuned structural basis of phospho-regulation of 4E-BP2:eIF4E binding (Fig. 6b). Apo np-4E-BP2 is pre-ordered, albeit transiently, in eIF4E-binding helical form[11]. Hierarchical phosphorylation of pT37 and pT46 shifts the part of the binding motif away from helical conformation toward β-strand and triggers the formation of the folded domain. Some of the binding residues are tucked into the β-sheet as β4, with β1 between it and the edge of the β-sheet. This partially sequesters the canonical binding motif, accounting for the weaker affinity of eIF4E for 2p-4E-BP2. However, some helical propensity remains in the C-terminal end of the binding motif. Previously, we found that the effects of C-IDR phosphorylation (pS65, pT70, pS83) required the presence of the folded domain[11]. Here, we have defined in detail the mechanism behind the three-stage attenuation of eIF4E binding that is the core of the phospho-regulation of 4E-BP2 inhibition. C-IDR phosphorylation further decreases the helical population

of the canonical and secondary eIF4E-binding sites. β1 and β4 are lengthened and have increased β-strand propensity. The canonical binding site residues are increasingly in the wrong conformation for binding, sequestered away from its binding partner, eIF4E.

**Phospho TPGGT hairpins turns are independently stable.** 4E-BP2 has a conserved stretch of residues near the S83 phospho-site, $^{83}$SPGT, that is similar to the TPGGT hairpin-forming motif, except that Ser replaces Thr and one of the glycine residues is missing (Supplementary Fig. 1a). There is no stable hairpin formed after S83 phosphorylation, based on the lack of additional downfield $^1$HN chemical shifts indicative of a phosphate hydrogen bond to the G85 amide proton. To test the independent stability of the phosphate-induced turn, we made a mutant with a glycine insertion between G85 and T86, and observed a third downfield $^1$HN chemical shift indicative of a phosphate hydrogen bond to the G85 amide proton. This new resonance confirms formation of a third hairpin within the disordered C-IDR upon phosphorylation ($^{83}$pSPGGT) (Fig. 7a) and the ability of pTPGGT/pSPGGT (or p[TS]PGGT) motifs to form stable hairpins independently of the 4E-BP2-folded domain.

The pTPGGT motif is conserved across species in 4E-BP1, 4E-BP2, and 4E-BP3 (Supplementary Fig. 1a)[11,23]. Given that 4E-BP2 folding requires hairpin formation, we tested the sequence specificity of hairpin formation by their effects on folding. Folding can be monitored in NMR spectra by the presence of the extremely downfield resonances diagnostic of hairpin formation and the other normally dispersed resonances reflecting stable structure. We find that despite the absolute conservation of phospho-threonine in the two hairpins, phospho-serine is also capable of forming a hairpin, both in the independent $^{83}$SPGGT motif introduced in the C-IDR as well as in a $^{37}$pSPGGT/$^{46}$pSPGGT 4E-BP2 double mutant (Supplementary Fig. 9a). However, the characteristic $^{39/48}$Gly peaks are not as down-field shifted in the double mutant as in the WT, suggesting that the latter is more stable. The pTPGGT is also conserved in the single isoform of 4E-BP found in invertebrates[34,35] with a notable exception of ticks, which have a TPGGS motif at the second hairpin (Supplementary Fig. 1a). Similarly, we find that the domain can fold in the context of select mutations to the Thr in the fifth position, with both $^{37}$pTPGGR and $^{37}$pTPGGA showing little disruption to the domain (Supplementary Fig. 9b, c). Mutations to the two glycine residues of the motif (the third and fourth positions) become increasingly more disruptive to the domain as the bulk of the substituted residue sidechain increases. $^{37}$pTPSGT and $^{37}$TPGLT 4E-BP2 show some peaks associated with the folded domain, but are missing many others (Supplementary Fig. 9d, e). Larger or more rigid β-branched residues, such as $^{37}$pTPIGT, $^{37}$pTPGPT, $^{37}$pTPGWT, $^{37}$pTPGVT (Supplementary Fig. 9f–i) or $^{37}$pTPVGT/$^{46}$pTPVGT (ref. [11]), are more disruptive to the domain. Even though these hairpin mutants destabilize the folded domain, most show evidence of some degree of hairpin formation based on downfield glycine chemical shifts indicative of the turn.

**Sequence requirements and frequency of TPGGT-like motifs.** Since a range of sequences can display the phospho-dependent hairpin structural behavior of 4E-BP's TPGGT motifs, we searched for the sequences compatible with the hairpin structure through bioinformatic assessment of similar hairpins observed in the PDB (described in "Methods" section). The primary and defining sequence feature associated with this structure is the glycine at the i + 3 position (TPGGT), while other residues are selective but can include a variety of residues at each position

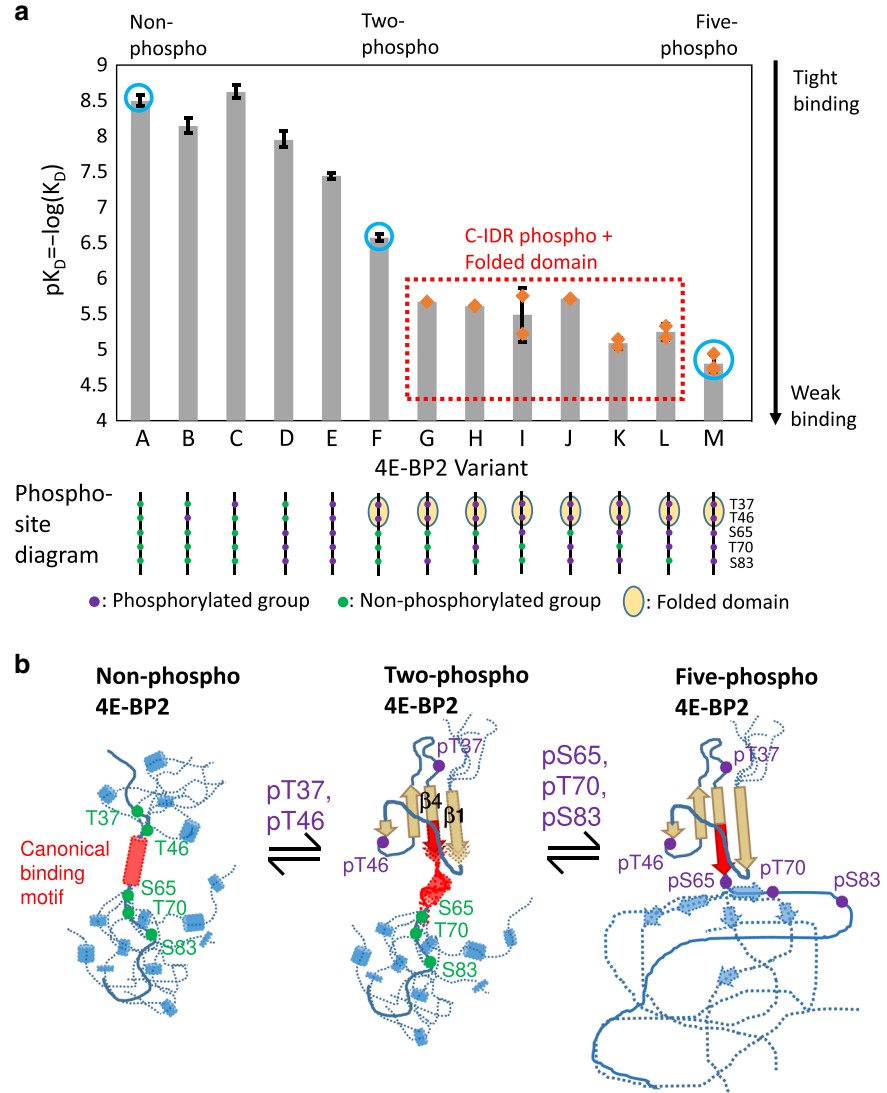

**Fig. 6 4E-BP2 phosphorylation tunes eIF4E binding. a** Phospho-dependent eIF4E-binding affinities. The eIF4E:4E-BP2-binding affinities for different 4E-BP2 variants are plotted as $pK_D = -\log(K_D)$ for (A) non-phospho, (B) pT46 (T37A/S65A/T70A/S83A), (C) pT37 (T46A/S65A/T70A/S83A), (D) C-IDR phospho pS65/pT70/pS83 (T37A/T46A), (E) 5-phospho Gly-to-Val double mutant: G39V/G48V pT36/pT46/pS65/pT70/pS83, (F) two-phospho pT37/pT46 (S65A/T70A/S83A), (G) pT37/pT46/pS83 (S65A/T70A), (H) pT37/pT46/pT70 (S65A/S83A), (I) pT37/pT46/pS65 (T70A/S83A), (J) pT37/pT46/pT70/pS83 (S65A), (K) pT37/pT46/pS65/pS83 (T70A), (L) pT37/pT46/pS65/pT70 (S83A), and (M) five-phospho pT37/pT46/pS65/pT70/pS83. The $K_D$ values and their uncertainties are from this work (G–M) or Bah et al.[11] (A–F, M) (see Supplementary Fig. 7) and were measured using ITC as the mean and standard deviation of sample repeats (see "Methods" section). Repeat data collections ($n = 3$ for 5p-4E-BP2; $n = 2$ for pT37/pT46/pT70, pT37/pT46/pS65, pT37/pT46/pT70/pS83, pT37/pT46/pS65/pS83, pT37/pT46/pS65/pT70; $n = 1$ for pT37/pT46/pS83 due to sample and data quality) were taken from different samples, some from the same preparation and others from different preparations. Individual $K_D$ values found for this paper (G–M) are indicated with orange diamonds. The affinities for np-4E-BP2, 2p-4E-BP2, and 5p-4E-BP2 are marked in the chart with blue circles. Schematic cartoons for each of the 4E-BP2 variants are shown at the bottom of the chart in which green circles indicate non-phosphorylated sites, purple circles are phosphorylated residues, and the tan oval represents the folded domain of 4E-BP2. **b** Schematic representation of three-stage phospho-regulation of 4E-BP2:eIF4E binding. Cylinders and arrow represent helices and β-strands, respectively. The disordered np-4E-BP2 and C-IDR of phospho 4E-BP2 is shown as multiple dashed lines with a single polypeptide highlighted as a solid line. Apo np-4E-BP2 is pre-ordered, albeit transiently, in eIF4E-binding helical form. The canonical binding motif is highlighted in red and the unphosphorylated site are marked as green spheres. pT37/pT46 (shown as purple spheres) phosphorylation triggers folded domain formation, tucking part of the binding residues into the β-sheet as β4, with β1 between it and the edge of the β-sheet. Some helical propensity remains in the C-terminal end of the binding motif. pS65/pT70/pS83 phosphorylation further shifts conformational propensities toward β-strand. β1 and β4 are lengthened and stabilized in the folded domain, sequestering the rest of the binding site.

(Fig. 8a, b). Confirming the validity of these statistics, hairpin mutations, which we tested in 4E-BP2, exhibited effects corresponding to their prevalence in similar turns within the PDB. Fold-compatible mutants, [37]pTPGGR and [37]TPGGA, are found in similar turns and fold-destabilizing mutants, [37]pTPIGT, [37]pTPGPT, [37]pTPGWT, and [37]pTPGVT, are excluded.

Beyond the enrichment of glycine at i + 3, we observe that T or S at i and P at i + 1 are also associated with an increased hairpin propensity, with 2640 unique [TS]P-containing hairpins found in the PDB. This suggests that the structurally similar turns found in constitutively folded proteins can include a proline-directed phospho-site, and that the stability of these folds could be

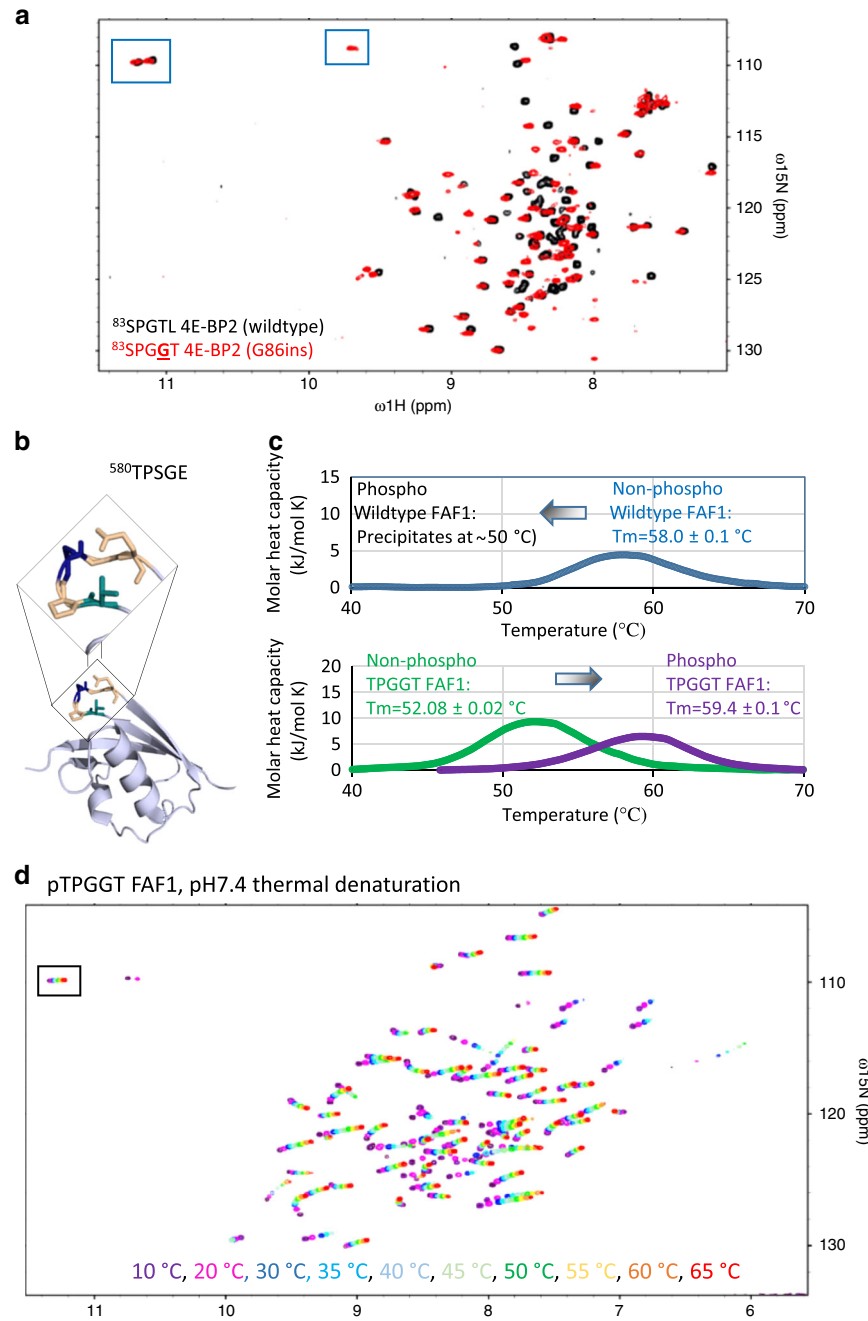

**Fig. 7 Independent stability of the pTPGGT motif. a** An extra hairpin can be added to the C-IDR of 5p-4E-BP2. 4E-BP2 has a pseudo-hairpin motif (83SPGT) in the C-IDR, which does not form a hairpin normally. After inserting a Gly between G85 and T86, a hairpin structure is formed, demonstrating the hairpin's independent stability. **b** Wildtype 580TPSGE motif of the ubiquitin-like domain of FAF1 (residues 570–650, PDB ID 3QX1) forms a turn structure similar to the hairpin. **c** DSC data show that phospho-regulatory stability of pTPGGT can be transplanted into the FAF1 domain. **d** NMR thermal denaturation of pTPGGT FAF1 domain. The hairpin Gly is resistant to denaturation at pH 7.4 in a manner strikingly reminiscent to phospho 4E-BP2 (see Fig. 1c).

phospho-regulated. We also note that there is a bias towards polar residues at the i + 4 position of the turn, whose sidechain would come in direct contact with any phosphate present at position i, and this includes enrichment for negatively charged residues which we would expect to be repelled by phosphate.

To test the importance of the i + 4 position and whether phospho-regulation of hairpin stability can occur in folded proteins, we first devised a position-specific scoring matrix (PSSM)-based scoring function for assessing the overall hairpin propensity of a given 11-mer sequence, which we made based on the amino acid frequencies observed in hairpin structures identified in the DSSP secondary structure database[36] (Fig. 8c and see "Methods" section). Next, we used the scoring function to score the 9764 unique 11-mer sequences with [TS]P*x*G (where *x* is any residue) found in the portion of the human proteome covered by the PhosphoSitePlus database[37]. For each individual protein, we then took the lowest scoring [TS]P*x*G containing 11-mer (lowest predicted hairpin propensity) and for the 6619 proteins containing at least one of these 11-mer sequences, we identified 418 proteins where the lowest [TS]P*x*G hairpin

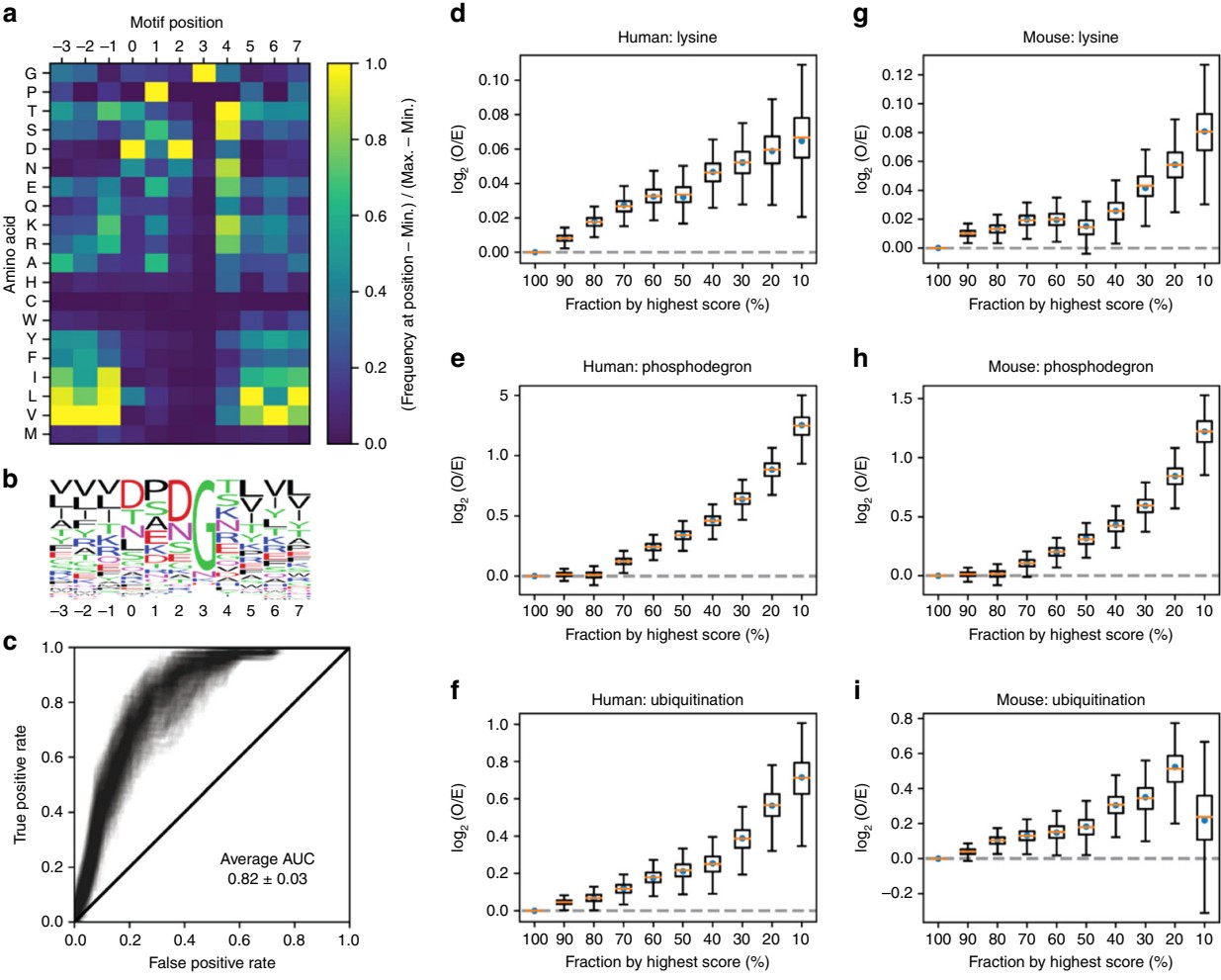

**Fig. 8 Bioinformatic analysis of hairpin sequence motifs. a** Sequence dependence of hairpin formation for the 11-residue linear motifs used to define the TPGGT hairpin turn, with motif position showing numbering relative to the TPGGT-associated positions of the turn. Colors show the relative hairpin propensity of each amino acid at each position, normalized to 1 (yellow) for the highest hairpin probability and 0 (black) for the lowest. **b** Sequence logo showing the underlying frequency distribution of panel **a**, from WebLogo[69]. **c** Receiver operating characteristic assessment of whether a hairpin turn derived PSSM can be used to predict hairpins for [TS]PxG containing sequences found in the PDB, gray lines showing true/false positive rates for 100 repeats of five-fold cross validation (500 curves in total) with training/testing data split based on a list of 3870 nonredundant PDB chains. **d–i** Testing the ability of the hairpin prediction score to predict ubiquitination-related properties, with **d**, **g** showing local lysine enrichment (30 residues on either side flanking the scored sequence), **e**, **h** showing overlap with two defined phosphodegron motifs (as described in "Methods" section), and **f**, **i** showing ubiquitination events observed within 30 residues in the PhosphoSitePlus database[37]. Enrichment values were calculated separately for two proteomes, human **d–f** and mouse **g–i**, with observed enrichment shown by blue circles and bootstrap analysis (sampling against 6619 human and 5084 mouse proteins) used to calculate 99% confidence intervals (shown by whiskers), interquartile range (shown by boxes), and medians (shown by orange lines).

propensity observed scored as highly as 4E-BP2, and 324 proteins with higher predicted propensity than any human 4E-BP sequence. We selected an example from the highest scoring 324 proteins that satisfies two criteria: (i) containing a negatively charged residue at position i + 4, and (ii) forming a constitutive hairpin as part of a folded domain. We identified the ubiquitin-like domain of Fas-associated factor 1 (FAF1, residues 570–650) (Fig. 7b) based on the hairpin turn score of its $^{580}$TPSGE sequence (it was in the top 269 out of 6619 human proteins in PhosphoSitePlus[37], when ranked by the lowest motif score observed per protein (Fig. 8a, b)). We then transplanted the TPGGT sequence into FAF1 and tested the effect on stability of phosphorylating both wildtype and TPGGT-modified domains.

The non-phospho $^{580}$TPSGE FAF1 domain has a single two-state transition with a $T_m$ of ~58 °C observed via DSC (Fig. 7c).

Phosphorylating T580 results in a loss of protein stability with most of the protein precipitated by ~50 °C, most likely due to the electrostatic repulsion between the negative charges of the phospho-group and the E584 carboxylate. Replacing the FAF1 $^{580}$TPSGE sequence with $^{580}$TPGGT, however, converts the effect of phosphorylation from destabilization to stabilization. The non-phospho TPGGT FAF1 domain has a $T_m$ of ~52 °C and phospho TPGGT FAF1 domain has a $T_m$ of ~59 °C, a 7 °C increase. Phosphorylation is a destabilization event for the wildtype FAF1 domain, with glutamic acid at the i + 4 position, but transplanting the TPGGT motif to FAF1 leads to stabilization upon phosphorylation. The pTPGGT FAF1 domain also has similar thermal denaturation behavior observed under NMR spectroscopy. At pH 7.4, the hairpin glycine residue is resistant to thermal denaturation, even up to 65 °C, in a strikingly similar manner to phosphorylated 4E-BP2 (Fig. 7d).

**Potential phospho-regulatory roles of TPGGT hairpin turns.**
The observation that proline-directed phosphorylation of TPGGT-like hairpins can both stabilize and destabilize folded domains suggests a more general role for modifying the stability of proteins in which these motifs are found. Phosphodegrons are short amino acid sequences that, when phosphorylated, route a protein for degradation[38]. Ubiquitin ligase Fbw7 recognizes two motifs that are similar to our TPGGT sequence: [LIVMP]$x$(0,2) TP$xx$E, which is singly phosphorylated at Thr, and [LIVMP]$x$(0,2)TP$xx$[ST], which is doubly phosphorylated[37,39,40] ('$x$(0.2)' indicates 0, 1, or 2 open spaces in the motif). To test the broader relevance of hairpin stability to degradation, we explored phosphodegron motif overlap and ubiquitination site annotation frequency as a function of our hairpin propensity score for the 7380 mouse and 9764 human [TS]P$x$G motifs found in the PhosphoSitePlus database[37]. We found that hairpin turn propensity correlates with three markers related to ubiquitination (Fig. 8d–i), including (i) the presence of local lysine residues (Fig. 8d, g), (ii) the likelihood of matching one of the aforementioned phosphodegron motifs (Fig. 8e, h) and (iii) the likelihood of ubiquitination being directly observed in the local sequence (Fig. 8f, i). These observations and results suggest that TPGGT-like hairpins could have a more general role in regulating degradation by modifying protein stability.

## Discussion

Why most native single domain proteins fold cooperatively is not clear; it could be a by-product of marginal stability, where there is no fitness benefit for any segment of a protein to be more stable than the whole, but it could also be adaptive, for example by making it easier to recognize misfolded proteins and target them for degradation. Here we demonstrate that stepwise, non-cooperative folding can have regulatory function, enabling the occupancy of a fold to be fine-tuned to regulate competition of processes involving either the folded or unfolded states. We previously demonstrated that C-IDR phosphorylation of 4E-BP2 reduces eIF4E-binding affinity by orders of magnitude, and our data strongly suggested that it does so by modulating the stability of the folded domain[11]. In this paper, we provide a detailed structural mechanism for the impact of the C-IDR phospho-sites on the folded domain to explain the full extent of phospho-regulation of translation initiation. The canonical eIF4E-binding region, which significantly samples helical structure in the non-phosphorylated state, is only partially converted to eIF4E-binding incompatible β-structure in the pT37/pT46 two-phospho state. C-IDR phosphorylation shifts the equilibrium away from helical to β-structure with full stabilization of the β-folded domain only with 5p-4E-BP2.

The previous model, in which phosphorylation leads to electrostatic repulsion between the 4E-BP and negative eIF4E-binding surfaces, is based on the proximity between S65 of 4E-BP1 and E70 eIF4E in crystal structures of eIF4E in complex with np-4E-BP1 fragments[18,21]. The pS65 site is positioned at the C-terminal end of the canonical binding helix, destabilizing the helix at physiological pH[10,22]. Consistent with this electrostatic effect on the helix dipole, isolated 4E-BP1 peptides containing the canonical eIF4E-binding site (residues 51–67) have a modest decrease in binding affinity upon S65 phosphorylation[41]. Our data argue that phosphorylation regulates 4E-BP2 binding by controlling its conformation, contrary to the electrostatic repulsion model. S65 is in a loop/turn region and the orientation of its sidechain varies between different complex structures[13,21]. Solution NMR on full-length 4E-BP2 reports on μs–ms motions of the C-IDR over a more extensive eIF4E-binding site than is apparent to X-ray crystallography on truncated 4E-BP proteins[8,17], a

dynamic interface supported by SAXS[23]. The dynamic nature of the complex complicates interpretation of binding energies from structural data, but our simplistic in silico modeling also supports the view that electrostatic repulsion is not the dominant mechanism by which S65 phosphorylation impacts binding. Disrupting the 5p-4E-BP2-folded domain (with a G39V/G48V double mutant) yields a protein that behaves more like tight-binding WT np-4E-BP2 than weaker-binding WT 5p-4E-BP2 ($K_D = 36.1 \pm 3.5$, $3.20 \pm 0.6$, and $12320 \pm 200$ nM, for five-phospho G39V/G48V, non-phospho WT, and five-phospho WT 4E-BP2, respectively)[11]. Together with the relatively tight ($K_D = 11.3 \pm 2.9$ nM) binding of pS65/pT70/pS83 3p-4E-BP2, also lacking a folded domain[11], these data provide very strong evidence that phosphorylation acts primarily by modulation of the fold.

The canonical and secondary eIF4E-binding motifs of 4E-BP2 are accessible to eIF4E binding, but the degree of accessibility depends on the particular 4E-BP2 phosphorylation state. Forming the folded domain after pT37/pT46 phosphorylation restricts access to the N-terminal part of the canonical-binding site, but the restriction is incomplete due to the non-cooperative folding of the domain, which offers some access to the motif. Phosphorylation at the C-IDR sites reinforces the β-sheet conformation of the domain, shifting it, along with the rest of the canonical site, further away from eIF4E-binding-competent conformations.

The two conserved pTPGGT motifs of 4E-BP2 are phosphorylated before the C-IDR sites, likely due to protection from various regulatory kinases when 4E-BP2 is in complex with eIF4E. However, in the absence of eIF4E, we show that the C-IDR sites are more susceptible to de-phosphorylation by lambda protein phosphatase. 2p-4E-BP1 is protected from ubiquitination and degradation only in complex with eIF4E, while in the absence of eIF4E, 2p-4E-BP2 is vulnerable to ubiquitination at K57 in the canonical-binding site[14], consistent with K57 existing in a dynamic equilibrium between helical and extended conformers in the two-phospho state. Interestingly, the free 5p-4E-BP1 is not ubiquitinated[14], consistent with the full stabilization of the folded domain incorporating the putative ubiquitination site K57 in a protected structure. Our data, including de-phosphorylation kinetics, in conjunction with the literature[14], suggest that the relative amounts of free 2p-4E-BP2 and 5p-4E-BP2 in cells are regulated by de-phosphorylation and ubiquitination based on the differential stabilities and protection afforded by the various phospho-states.

The TPGG[ST] sequences are conserved features of all three 4E-BP isoforms, with high positional sequence conservation throughout the proteins (Supplementary Fig. 1a), fairly unusual for IDPs[42]. This sequence conservation is constrained by (i) the structural requirements of the folded domain, (ii) the need to maintain specific eIF4E:4E-BP versus eIF4E:eIF4G binding and (iii) binding to kinases, phosphatases, and ubiquitin ligases. The sequence gives rise to a protein for which stepped phosphorylation enables tuned stabilization of a non-cooperatively folded domain, controlling exposure of the canonical eIF4E-binding helix and ultimately competition for eIF4E binding to eIF4G, required for translation initiation. This knowledge of the underlying mechanism of stability of the various phospho-states of 4E-BPs and other similar hairpin-containing sequences, including their vulnerabilities to ubiquitin-dependent degradation, may provide valuable insights for controlling a variety of biological processes.

## Methods

**Construct and sample generation.** ACGT Corp (Toronto, ON) synthesized mutagenesis primers (see Supplementary Methods) and performed DNA sequence analysis for the wild type and mutants used in this study. Site-directed mutagenesis

was performed in-house using standard Quikchange protocols (Agilent) or out-sourced to either Genscript Inc or ACGT Corp. Invitrogen GeneArt (Life-technologies) synthesized a codon-optimized cDNA of the ubiquitin-like domain of FAF1 (residues 570–650) and subcloned it into a pET SUMO-vector, generating an expression vector similar to those of the eIF4E and 4E-BP2 constructs. The proteins were expressed in BL21-CodonPlus (DE3)-RIPL $E. coli$ cells (Agilent Technologies) in Luria Broth for unlabeled samples and uniformly $^{15}N$ or $^{15}N/^{13}C$-labeled M9 for NMR samples. The cells were cultured at 37 °C until $OD_{600}$~0.6–0.8, induced with IPTG, and overexpressed at 16 °C for 16–20 h. Cell lysates were purified using a nickel–nitrilotriacetic acid (Ni–NTA) column. The SUMO solubility tag was cleaved by 30-min incubation with Ulp1 and then removed by the Ni–NTA column. The proteins were purified to homogeneity using S75 HiLoad gel filtration columns (GE Healthcare). Activated His-tagged Erk2 for phosphorylation was expressed and purified using a protocol and plasmid co-expressing Erk2 and MEK1 obtained from Attila Remenyi at Eötvös Loránd University. 4E-BP2 and FAF1 were phosphorylated using Erk2 using a dialysis technique. Each ~50 ml phosphorylation reaction was made up of phosphorylation buffer (50 mM Tris pH 7.5 at 25 °C, 1 mM EGTA, 2 mM DTT, 20 mM $MgCl_2$, and 10 mM ATP) containing ~20 μM 4E-BP2 and ~5 μM Erk2 in a dialysis bag. The dialysis bag was placed in 1 l of phosphorylation buffer and phosphorylation proceeded for 1–3 days with stirring, then the reaction was stopped by removing the kinase with a Ni–NTA column. Flow-through and wash fractions of phosphorylated protein were purified via gel filtration.

**Alignment of 4E-BP sequences.** The 4E-BP sequences were extracted from the Uniprot Database[43] using the database's BLASTP server[44]. The sequences were aligned with CLUSTAL OMEGA[45].

**In silico-bound state energy minimization.** Structural studies of eIF4E bound to 4E-BP fragments provide a binding-energy hypothesis for the mechanism of hierarchical phosphorylation, where phosphosite S65's close bound-state proximity to a glutamine sidechain suggests that phosphorylation of S65 may weaken binding by electrostatic repulsion[18,21]. We tested whether the structural context of S65 confirms electrostatic repulsion in silico by doing energy minimization analysis of the bound structure with and without phosphate and found no detectable energetic difference between states (Supplementary Fig. 1c), demonstrating that, while electrostatic change could potentially have some effect on binding, the effect is not captured by coulombic (electrostatic) interactions modeled in the bound state. The electrostatic effects of phosphorylation on bound state energy were modeled using Rosetta version 3.10[46] and its energy function[47] using the default command line for the relax application (relax -in:file:s 4ued.pdb). A phosphate group was added to S65 by modifying the input file, changing the amino acid code for S65 from SER to SEP (phospho-serine) which triggers Rosetta to build in the missing phosphate atoms. Two different net charges were tested for the phosphate group: the default for Rosetta 3.10 (−1.17) and one closer to the expected charge under physiological conditions (−1.88). This second charge was defined by replacing the partial charge values in Rosetta database file "/database/chemical/residue_type_sets/fa_standard/ patches/ser_phosphorylated.txt" with values taken from Steinbrecher et al.[48], with oxygen O1P, O2P, and O3P partial charges changed from −0.78 to −0.98 and phosphate partial charge changed from 1.5 to 1.39.

Rosetta score minimization against the energy function[47] was used to gauge the potential for electrostatic repulsion in bound state structures of S65 phosphorylation. Five thousand minimized structures were generated for each state starting from atom coordinates in the human eIF4E:4E-BP1 structure (PDB ID 4UED). Non-phospho S65 uses the default minimization protocol and the phosphorylation states use the same protocol but with the addition of phosphate groups, with net charge tested at around both −1 and −2. Supplementary Fig. 1b shows an overlay of the best scoring structure from each set superimposed to the PDB structure, with eIF4E in gray for all sets, and 4E-BP colored in gray for PDB:4UED, cyan for the best scoring non-phospho model, magenta for phosphate charge −1, and yellow for phosphate charge −2, demonstrating that phosphorylation does not require perturbation of the overall structure, and Supplementary Fig. 1c shows that the minimization energy distribution is indistinguishable between states, even with different net charges used on the phosphate. The boxes in Supplementary Fig. 1c show the interquartile range of energy scores, with a white dot labeling the median.

**Differential scanning calorimetry.** All experiments were performed on a Nano DSC equipped with an Autosampler (TA Instruments) with a heating rate of 1 K/ min using protein concentrations ranging from 1 to 10 mg/mL. A Cys-less (C35S/ C73S) 4E-BP2 construct was used to optimize the DSC signal. Data were collected and analyzed using the DSCrun and NanoAnalyze programs provided with the instrument. $N = 3$ data sets were collected at each condition using different samples, some from the same preparation.

**NMR chemical shift analysis.** NMR experiments were collected on 500, 600, and 800 MHz Varian NMR spectrometers (the field for each experiment is specified in the below sections). The data were processed with nmrPipe[49], and Sparky[50] was used for spectra viewing, overlays, and assignment transfer. Sensitivity-enhanced

$^{15}N$–$^1H$ HSQC spectra[51] were collected for chemical shift analysis of denaturation, phosphorylation, and mutational data. When analyzing NMR data, peaks that were overlapped with other peaks in the NMR spectra were eliminated from analysis, as were peaks near the background noise level of the spectra. Peaks that shifted too far from their original positions to be assigned with confidence (due to mutation or de-phosphorylation) are also excluded from explicit analysis, though their location was noted on the 4E-BP2 protein structure.

Changes in chemical shifts, $\Delta\omega$, due to denaturation, mutation, or other modifications were calculated using Eq. (1).

$$\Delta\omega = \sqrt{\Delta\omega_{1H}^2 + 0.154^2 * \Delta\omega_{15N}^2} \tag{1}$$

Chemical denaturation of 5p-4E-BP2 was done using GdmCl and urea. Both sets of denaturation experiments were performed at 20 °C and pH 6.0 with samples in phosphate buffer (30 mM $Na_2HPO_4$, 100 mM NaCl, 2 mM DTT, 1 mM EDTA, pH 6.0) with various concentration of denaturant added. The pH was adjusted back to pH 6.0 with HCl and NaOH after the addition of GdmCl. The urea concentrations shown are only approximate, due to water and degradation products. $^{15}N$–$^1H$ NMR sensitivity enhanced HSQC spectra[51] were collected at 500 MHz and the chemical shifts are referenced to water.

To investigate the pH-dependence of the stability of the folded domain of 5p-4E-BP2, samples with pH values between 2.2 and 8.0 were made with McIlvaine (phosphate–citrate) buffer[52] with 2 mM DTT. The pH 1.0 samples were in 30 mM trichloroacetic acid (TCA) with 2 mM DTT (pH adjusted), since this pH is out of McIlvaine buffer's buffering range. Only the data for samples in McIlvaine buffer were used to fit the pH titration data, and their chemical shifts were referenced to DSS. At the pH values investigated, titration of His, phosphoThr (pThr), Asp, and Glu residues in the phospho domain is expected to be observed. Supplementary Fig. 2a shows a group of folded domain residues shifted toward random coil values (~8.4 ppm). Many show curvatures indicating multistate transitions. To find the $pK_a$ values and pH-dependent free energies of unfolding, we fit the two-state and three-state protonation models using bootstrap minimization with a 1000 iterations implemented with python and numpy. The fitted parameters are given as mean ± standard deviation.

Two-state protonation model:

$$A^- + H^+ \rightleftharpoons BH$$
$$\omega_{obs} = \frac{\omega_1 + K^*\omega_2}{1 + K}$$
$$K = 10^{-(pH-pK_a)} \tag{2}$$

$$\Delta G_{pH}^{unfold} = -RT\ln\left(10^{-(pH-pK_a)}\right) = +RT\ln(10)(pH - pK_a) \tag{3}$$

Three state protonation model: $A^{2-} + H^+ \rightleftharpoons BH^- + H^+ \rightleftharpoons CH_2$

$$\omega_{obs} = \frac{\omega_1 + K1^*\omega_2 + K1^*K2^*\omega_3}{1 + K1 + K1^*K2}$$
$$K1 = 10^{-(pH-pK_{a1})}$$
$$K2 = 10^{-(pH-pK_{a2})} \tag{4}$$

$$\Delta G1_{pH}^{unfold} = +RT\ln(10)(pH - pK_{a1})$$
$$\Delta G2_{pH}^{unfold} = +RT\ln(10)(pH - pK_{a2}) \tag{5}$$

For the pH titration data, the weighted sum of the $^1H$ and $^{15}N$ chemical shifts ($\omega_{obs} = \omega^1H + 0.154*\omega^{15}N$) was used to reduce noise.

The chemical shift behavior was used to distinguish between residues affected by acid denaturation of the folded domain and solvent-exposed residues affected by buffer conditions. Residues that shift toward random coil are likely to be affected by denaturation (Supplementary Fig. 2a). The $pK_a$ values for the residues affected by acid denaturation, referred to in the text as apparent $pK_a$, varied between 3.62 and 5.40, consistent with heterogeneity in pH-dependent unfolding free energy (Fig. 2a, Supplementary Table 1). Sidechains that titrate independently of denaturation are also observed for some residues. For example, pH titration data for residue Q29 fit to a three-state model. The chemical shift changes at higher pH values move toward random coil and probably reflect denaturation (Supplementary Fig. 2a). The lower pH titration, however, moves in the opposite direction and has $pK_{a2} = 2.3 \pm 0.4$. The direction is consistent with proximity to a residue that is becoming less negative at lower pH, probably D26 or D33. H32 has two titrations with $pK_a$ values at pH $6.0 \pm 0.2$ and $2.6 \pm 0.1$, most likely due to the H32 and D33 sidechains. The lower pH data points for D33 were excluded from analysis due to peak overlap. However, H32 titrates with a $pK_a = 5.9 \pm 0.1$. Y34 was also affected by H32 with a $pK_{a1} = 6.0 \pm 0.3$. T45 fits the three-state model with the lower pH titration moving away from random coil, probably also corresponding to the titration of a nearby acidic residue.

**Analysis of NMR thermal denaturation data.** $^{15}N$–$^1H$ HSQC thermal denaturation spectra were collected for 5p-4E-BP2 samples in pH 1.0, 2.6, 4.0, 5.0, 6.0, 7.0, and 7.4 buffers at temperatures between 5 and 70 °C. 70 °C was the operational temperature limit for the 500 MHz spectrometer and its probe. As with the pH titration data, samples at pH 1.0 were in TCA buffer. The rest were in McIlvaine buffer and the data from only these samples were used in the analysis of thermal

denaturation. As a comparison, HSQC spectra of disordered np-4E-BP2 in pH 5.0 McIlvaine buffer were collected at temperatures between 5 and 70 °C.

For the 5p-4E-BP2 data, many of the folded domain peaks do not reach random coil values, even at pH 1.0 and 70 °C (Figs. 1c and 2c), so we could not obtain a complete thermal denaturation curve. We did, however, develop a way to indirectly estimate the thermal denaturation energies. The folded domain is less stable at low pH and is more sensitive to thermal denaturation. Based on this observation, we propose a three-state model, in which protonation is required for thermal denaturation.

Three state thermal model:

$$A^- + H^+ \rightleftharpoons BH \rightleftharpoons CH$$
$$\omega_{obs} = \frac{\omega_1 + K_{pH}*\omega_2 + K_{pH}*K_{thermal}*\omega_3}{1 + K_{pH} + K_{pH}*K_{thermal}} \quad (6)$$

$$K_{pH} = 10^{-(pH - pK_a)}$$
$$K_{thermal} = e^{-\Delta G(T)/RT} \quad (7)$$
$$\Delta G(T) = \Delta G° - \Delta S°*(T - T°) + \Delta C_p*\left(T - T° - T*\ln\left(\frac{T}{T°}\right)\right)$$

Equation (7), specifically the expression for $\Delta G(T)$, has been used previously to model the thermodynamics of protein folding[53]. As for acid denaturation, we take the pH-independent free energy to be the unfolding energy. For this analysis, the reference temperature for the standard free energy, entropy, and change in heat capacity (denoted with °) is 20 °C, matching the pH titration data.

Since we could not obtain a complete thermal denaturation curve, we instead calculated apparent $pK_a'$ values at different temperatures. In order to find the apparent $pK_a'$ at each available temperature, we recast Eq. (6) in two-state form.

Generic, target two-state model:

$$\omega' = \frac{\omega'_1 + K'_{pH}*\omega'_2}{1 + K'_{pH}} \quad (8)$$

Converting Eq. (6) into two-state form:

$$\omega_{obs} = \frac{\omega_1 + K_{pH}*(\omega_2 + K_{thermal}*\omega_3)}{1 + K_{pH}*(1 + K_{thermal})}$$

Assuming $\omega_2 \sim \omega_3 = \omega'_2$ and casting this into the two-state form:

$$\frac{\omega_1 + K_{pH}*(1 + K_{thermal})*\omega'_2}{1 + K_{pH}*(1 + K_{thermal})} = \frac{\omega'_1 + K'_{pH}*\omega'_2}{1 + K'_{pH}} \quad (9)$$

We now have a relation between $pK_a'$, the apparent $pK_a$ that is affected by thermal denaturation, and the actual $pK_a$:

$$K'_{pH} = K_{pH}*(1 + K_{thermal})$$
$$10^{-(pH - pK'_a)} = 10^{-(pH - pK_a)}*(1 + K_{thermal}) \quad (10)$$
$$10^{+(pK'_a - pK_a)} - 1 = K_{thermal} = e^{-\Delta G(T)/RT}$$

For this model, since $K_{thermal} \geq 0$, then $pK_a' \geq pK_a$. If $pK_a$ is known (see below), then $\Delta G(T)$ can be calculated using $pK_a'(T)$ for that temperature. The $\Delta G(T)$ values can then be fitted to Eq. (7) for the standard free energy, energy, and change of heat capacity. The errors for parameters extracted from curve fitting were estimated using bootstrap analysis with a 1000 iterations or by propagation of error. The fitted parameters are given as mean ± standard deviation.

To illustrate the method, here we focus on the analysis of G39 thermal denaturation data. The pH titrations for hairpin glycines G39 and G48 are shown in Supplementary Fig. 3b. Each of the different colors represent a pH titration at a specific temperature. In order to find the thermal denaturation energies, $\Delta G(T)$, we first estimate the true $pK_a$. To increase the number of data points to fit, we fit the $^1H$ and $^{15}N$ peaks simultaneously to Eq. (9).

$$\omega_{obs}^{1H} = \frac{\omega_1^{1H'} + K'_{pH}\omega_2^{1H'}}{1 + K'_{pH}}$$

$$\omega_{obs}^{15N} = \frac{\omega_1^{15N'} + K'_{pH}\omega_2^{15N'}}{1 + K'_{pH}}$$

Supmmentary Fig. 3c and Supplementary Data 1 shows the fitted apparent $pK_a'$ value at each temperature for G39. For G39, the value of $pK_a'$ plateaued at its minimum at the three lowest temperatures. The average and standard deviation of these three values were used to estimate the true $pK_a$ and its error: 5.19 ± 0.02. Then we use Eq. (10) to calculate $K_{thermal}$, then $\Delta G(T)$. After $\Delta G(T)$ values are calculated, Eq. (7) is fitted to calculate $\Delta G^0$, $\Delta S^0$, and $\Delta Cp$ (Supplementary Fig. 3d).

In general, $pK_a$ is estimated for other residues by averaging the $pK_a'$ points in the low temperature plateau. If this method does not work, we fit $pK_a'$ vs. temperature to linear, cubic, and quadratic curves to roughly estimate the $T = 0$ °C intercept for each and then average the intercepts. This second method results in much larger uncertainties than the first method. Most folded domain residues have minimal pH-independent free energy at 20 °C, i.e. $\Delta G°$ (Supplementary Fig. 3e, Supplementary Table 2). The exceptions are residues near or at the hairpins (pT37, G39, pT46, and G48), which have significantly more unfolding free energy. Theses

residues, while not as obviously affected by three-state pH titration events like Q29 and H32, do have chemical shift change trajectories that deviate from the straight line expected for two-state pH titrations (Supplementary Fig. 2a). Each hairpin has a hydrogen bond between a phosphoThr phosphate group and the (i + 2) glycine's amide group (pT37/G39, pT46/G48)[11]. This strongly implies that the pH-independent energy estimated here characterizes the strength of the hydrogen bonds in the hairpins. The presence of the additional thermal transition also explains why these residues, while not as obviously affected by three-state pH titrations such as Q29 and H32, have chemical shift trajectories that deviate from the straight line expected for two-state pH titrations.

The observed apparent $pK_a'$ values increase with temperature (e.g. Supplementary Fig. 3c, Supplementary Data 1). Another possible cause of this phenomenon is the temperature dependence of the buffer $pK_a$. While this possibility cannot be completely eliminated, it should be noted that the temperature dependence of the $pK_a$ values for phosphate buffers, such as the ones used here, is negative: $dpK_a/dT = -0.0028$[54]. The hairpin residues associated with pT37 and pT46 have significant thermal denaturation free energies and their apparent $pK_a'$ increased with increasing temperature, the opposite direction expected for the $pK_a$ shift of a phosphate buffer.

**NMR assignment transfer and secondary structure propensities.** The $H_N$, N, $H_{alpha}$, C', $C_{alpha}$, and $C_{beta}$ peak assignments were found using standard triple resonance experiments and experiments specialized for proline-enriched/IDP proteins[51,55–57] for pH 3.0 and 7.4 5p-4E-BP2, as well as 3p-4E-BP2, as well as 4p-4E-BP2 at pH 7.4, aided by the previously published pH 6.0 5p-4E-BP2 assignments[11]. The secondary structure propensity program SSP[28] and each set of $H_{alpha}$, C', $C_{alpha}$, and $C_{beta}$ chemical shifts were used to quantify secondary structure propensities.

**NMR paramagnetic resonance enhancement (PRE) data.** PRE samples for 2p-4E-BP2 (S65A/T70A/S83A) and 5p-4E-BP2 were constructed using Cys-less versions of both proteins (C35S/C73S). The H32C mutation was made to both constructs with standard Quikchange protocols (Agilent). The proteins were purified as before, labeled with TEMPO-maleimide spin label (Toronto Research Chemicals), phosphorylated (as before), and then given a final purification with gel filtration chromatography. Two PRE samples were made for each protein, one with the spin label oxidized and the other with the tag reduced. To oxidize the samples, five-fold excess of TEMPOL (Toronto Research Chemicals) was mixed with the sample. The reduced samples were made by mixing 1 mM ascorbic acid and the sample. Both the oxidized and reduced samples were incubated overnight with shaking, then dialyzed three times into NMR buffer. PRE data were collected using sensitivity-enhanced HSQC spectra[51] with a recycle delay (d1) of 5 s. For denatured or disordered residues, this method is considered more sensitive to PRE attenuation than the T2 PRE measurement techniques[58]. Separate HSQC spectra were collected for the oxidized and reduced samples at 500 MHz and 20 °C. The signal attenuation for each residue was estimated using the ratio of peak volume for the oxidized sample to that of the reduced sample, $V(ox)/V(red)$. The ratio is proportional to the distance separating the TEMPO tag from the backbone amide group. Therefore, signal attenuation ($V(ox)/V(red) < 1$) indicates that the TEMPO tag is proximal to a residue. The uncertainty in the signal attenuation is based on the propagation of the fit heights of the peak volumes.

**Hairpin motif mutant 4E-BP2 HSQC spectra.** $^{15}N–^1H$ HSQC spectra of hairpin motif mutants were collected at 20 or 25 °C on either a 500 or 600 MHz NMR spectrometer. The wildtype 5p-4E-BP2 spectra shown in Supplementary Fig. 9 were recorded under matching temperature, pH, and magnetic field conditions.

**Single-molecule FRET.** Single-molecule FRET (smFRET) experiments used two constructs, H32C/S91C with the native cysteines mutated to serine (i.e. C35S/C73S) on the 5p-4E-BP2 and the 2p-4E-BP2 (pT37/pT46 S65A/T70A/S83A) background. The smFRET samples were labeled with thiol-reactive dyes, i.e., Alexa 488 (A488) as donor and Alexa 647 (A647) as acceptor (ThermoFisher Scientific, Canada). For the labeling reaction, the dyes were added to a 50 μL solution of 100 μM protein at a A488:A647:protein molar ratio of 1.3:3:1 in the presence of tris(2-carboxyethyl)phosphine (TCEP) at a 10× molar excess to the protein. All maleimide–cysteine coupling reactions were performed in PBS buffer at pH 7.4. Oxygen was removed by flushing the sample with argon gas in a desiccator for 5 min. The vial was capped tightly and shaken gently for 3 h at room temperature. The excess dye was removed by size-exclusion chromatography using Sephadex G-50 gels (G5080, Sigma Aldrich) in a BioLogic LP system (731-8300, Bio-Rad).

In order to estimate the intensity correction factor ($\gamma$) for smFRET analysis[59], donor-only and acceptor-only labeled protein samples were prepared using a similar protocol. Using the free dyes as reference, the fluorescence quantum yield (QY) of A488 attached to 4E-BP2 was estimated to be 0.76, 0.81, and 0.83 for non-phospho, two-phospho, and five-phospho conditions, respectively. Under similar conditions, the 4E-BP2-bound A647 had QY values of 0.32, 0.34 and 0.35, respectively. The error margins of the estimated QY values are around 5–10%.

All samples were diluted to concentrations of 20–50 pM for smFRET burst experiments. For a typical experiment, a sample solution of ~30 μL was dropped on

the surface of plasma-cleaned glass coverslip. To prevent protein adsorption, the coverslip was coated with bovine serum albumin (BSA) (15260-037, ThermoFisher Scientific) as described previously[60,61], and 0.005% (v/v) Tween-20 (P2287, Sigma-Aldrich) was added to the solution. All experiments were performed at 20 °C.

smFRET measurements were performed on a custom-built multiparameter fluorescence microscope that was described in detail elsewhere[60,61]. The sample is excited at 480 nm by femtosecond pulses and the fluorescence signal is filtered and then recorded by sensitive avalanche photodiodes, which are fed into a time-correlated counting module (PicoHarp300, PicoQuant). A custom-written Matlab code identified and analyzed dual-color fluorescence bursts, and displayed the results as FRET efficiency histograms.

The FRET efficiency ($E$) was calculated based on the number of detected photons in both donor ($I_D$) and acceptor ($I_A$) channels in each single-molecule intensity burst[59]

$$E = \frac{I_A}{I_A + \gamma I_D},\tag{11}$$

where $\gamma$ is the correction factor for differences in detection efficiencies of donor and acceptor channels and quantum yields of the dyes. In addition, corrections were applied on both $I_D$ and $I_A$ to subtract background and spectral crosstalk. The dyes retained considerable rotational freedom under all conditions investigated and therefore the reported FRET $E$ values are good measures of proximity distances and can be directly compared among various conditions tested[33,62]. $E$ was calculated for each detected burst, and all values obtained from a sample were used to build a histogram. Gaussian fits to the data were used to indicate the locations of the peak FRET efficiency for the non-zero Gaussian. The average FRET values have an error margin of ±0.01 (the fitting error).

**Fluorescence correlation spectroscopy.** FCS experiments were performed on a custom-built instrument following a protocol described in detail elsewhere[63]. FCS experiments used constructs in the Cys-less (C35S/C73S) 4E-BP2 background with one of six Cys mutations: C0 (inserted before first residue), S14C, C35, C73, S91C, and C121 (inserted after sequence). Each single cysteine protein was labeled with A488 by adding the A488 fluorophores at a dye:protein molar ratio of 3:1 to a 50 μl solution of 100 μM protein. The rest of the preparation proceeded as described above for the smFRET samples. Non-phospho and five-phospho samples were diluted to concentrations of 1–10 nM in pH 3 and pH 7.4 buffers and FCS data was acquired for all four conditions (Supplementary Fig. 6). The coverslips were prepared as before and all experiments were performed at 20 °C. The laser excites the sample at intensities of ~5 kW/cm$^2$ in FCS measurements. The experimental correlation decay curves were fit to a to the typical model of molecular diffusion and dye photophysics, as given by Eq. (12):[61]

$$G(\tau) = \frac{1}{N_{eff}} \left(1 + \frac{\tau}{\tau_d}\right)^{-1} \left(1 + \frac{\tau}{s^2 \tau_d}\right)^{-0.5} \sum_i \left(1 + a_i e^{-\tau/\tau_d}\right)\tag{12}$$

In Eq. (12), $N_{eff}$ is the average number of molecules in the confocal detection volume; $s$ is the ratio between the axial and the lateral radii of the detection ellipsoid ($z_0/w_0$); $\tau_d$ is the diffusion time through the confocal volume, which is related to the diffusion coefficient ($w_0^2 = 4D\tau_d$) and the hydrodynamic radius, $R_H$, of the molecule via the Stokes–Einstein equation[64]:

$$R_H = \frac{k_B T}{6\pi\eta D}\tag{13}$$

In addition, $a_t$ and $\tau_t$ are the amplitude and the lifetime of the triplet (dark) state of the fluorophore, which also causes fluorescence intensity fluctuations[65]. Prior to each set of FCS measurements, a sample of Rhodamine 110 dye was used to characterize the geometrical parameters of the confocal detection volume[64].

**ITC affinity measurements for phospho 4E-BP2:eIF4E binding.** The ITC instrument (MicroCal ITC200) and data collection methods (using Auto-ITC200, an Origin software package from MicroCal) are detailed previously[11]. The ITC data for each construct is shown in Supplementary Fig. 8. The $K_D$ values and their uncertainties are from this work or Bah et al.[11] (see Supplementary Fig. 7). Repeat data collections ($n = 3$ for 5p-4E-BP2; $n = 2$ for pT37/pT46/pT70, pT37/pT46/pS65, pT37/pT46/pT70/pS83, pT37/pT46/pS65/pS83, pT37/pT46/pS65/pT70; $n = 1$ for pT37/pT46/pS83 due to sample and data quality) were taken from different samples, some from the same preparation and others from different preparations. The $K_d$ values and their uncertainties were estimated as the mean and standard deviations. The WT 5p-4E-BP2 $K_d$ value found for this article is comparable to that from Bah et al.[11].

**Bioinformatic assessment of pTPGGT structure and sequences.** In order to identify sequences compatible with the hairpin structure found in the folded domain of 4E-BP2 (PDB ID 2m × 4), we first defined a secondary structure and hydrogen bonding-based definition of the hairpin that could be categorized using the DSSP secondary structure assignments[36]. Potential turns were identified using an 11-mer DSSP secondary structure assignment of '?'-'E'-'E'-' '-'T'-'T'-' '-' '-'E'-'E'-'?' to identify potential turns, where 'E' is extended conformation, 'T' is a turn, '?' means any class, and a blank (' ') means any class but E, T, H (alpha-helix), or

G ($3_{10}$ helix). The following set of hydrogen bonds were then used to confirm that they match the hairpins observed in the 4E-BP2 structure: 2-O to 10-NH, 2-NH to 10-O, 4-O to 7-NH, and 4-NH to 8-O.

We then downloaded a comprehensive set of 153,773 DSSP files from the DSSP database (rsync://rsync.cmbi.umcn.nl/dssp). Since many structures in the PDB are redundant, we chose to use this redundancy to prioritize hairpins that are consistently observed even when solved multiple times, first identifying 49,938 non-redundant protein chains by using the PISCES PDB culling server[66] (resolution threshold of 3 Å, $r$-factor threshold of 1.0, and a sequence identity threshold of 90%) and then by clustering all of the culled redundant chains with their corresponding non-redundant exemplar. For each one of these sets we then collated the complete list of unique 11-mer sequences observed, and only accepted hairpins when they satisfy the DSSP criteria in at least 50% of the chains they are observed in over the cluster. 8917 cluster-unique hairpin observations were considered this way and 1136 were rejected for being inconsistent across multiple protein structures, leaving $N = 7781$ confident hairpin-containing 11-mers identified over a set of 5611 nonredundant protein chains.

To make a score function for hairpin prediction we generated a PSSM[67,68] for these 11-residue windows by taking the log2 probability of each amino acid at each position divided by the overall amino acid probability, with probabilities factoring in a small pseudocount of 0.1. For scoring 11-mers we then take the sum of log scores over each position. Positional frequencies used in generating PSSMs are diagrammed in Fig. 8a, b.

In order to assess the ability of this PSSM score to predict hairpins in the [TS] P$x$G motifs being studied we then went back through our protein clusters and identified a set of 5015 11-mers which match the sequence filter "$xxx$[TS] P$x$G$xxxx$", 858 of which were found to also contain hairpins structures, for a negative/positive ratio of ~4.8 to 1. Unlike the negative data, the positive data overlap with the sequences used to define the PSSM. This is because the positive data come from 570 of the 5611 nonredundant protein clusters identified previously. To assess the method's ability to score hairpins that it has not been trained on, we used five-fold cross-validation repeated 100 times, with training and testing sets defined by splitting the list of nonredundant [TS]P$x$G containing PDB chains used, and then by removing test set hairpin sequences from the pool used to generate the PSSM. ROC-AUC for these tests averaged to 0.83 ± 0.03 (Fig. 8c), which is reasonable for the purpose of predicting enrichment.

Correlations between hairpin propensity scores for [TS]P$x$G motifs and proteomic data were tested by scoring unique $xxx$[TS]P$x$G$xxxx$ motifs containing 11-mer sequences found in the PhosphoSitePlus[37] database, independently for both the human and mouse proteomes. To minimize the influence of highly repetitive sequences, 11-mers which show up multiple times in the same protein were reduced to the most C-terminal example, leaving 8015 human 11-mers from 6619 proteins and 6826 mouse 11-mers from 5084 proteins. We assessed three properties related to ubiquitination using the following definitions: (1) direct ubiquitination, measured by the observation of ubiquitination events in the PhosphoSitePlus database that in the primary sequence are within 30 residues of the 11-mer in question, (2) presence of local lysine residues, measured by the occurrence of lysine in the 30 residues preceding the 11-mer sequence or the 30 residues after it (representing local regions which are not scored directly by the PSSM score), and (3) overlap with phosphodegron motifs [LIVMP]$x$(0.2)TP$xx$E and [LIVMP]$x$(0.2)TP$xx$[ST][37,39,40].

Enrichment was measured by comparing log2($O/E$) for the frequency of observed features in high scoring sets of sequences divided by the frequency over the full set. Significance of enrichment was tested using bootstrap analysis, sampling the 6619 human and 5084 mouse proteins (with replacement) 10,000 times to calculate the 99% confidence intervals. Doing this across multiple PSSM score thresholds shows that there is significant enrichment ($p < 0.01$) for the top 20% by PSSM score over all three properties for both human and mouse (Fig. 8d–i), and for 5 out of 6 of the top 10%.

**Reporting summary.** Further information on research design is available in the Nature Research Reporting Summary linked to this article.

## Data availability

The source data (NMR chemical shift and peak intensity data) underlying Figs. 2a, b, 3, 4, 5, Supplementary Figs. 1a, 3, and 4, as well as Supplementary Tables 1 and 2 are provided in the source data file. The Hα, Cα, Cβ, CO, Hn, and N chemical shift values used to calculate secondary structural propensities are also found in the BMRB with the ID 50253. All other data are available from the corresponding author on reasonable request. Source data are provided with this paper.

## Code availability

Data analysis scripts are available from GitHub repositories. Python scripts for image generation and NMR chemical shift analysis are found in the directory jdawson_scripts at https://github.com/jenniferdawson/4EBP2_nmr/. Python scripts for bioinformatics analysis are available from 'https://github.com/jenniferdawson/Robert_Vernon_Bioinformatics_HairpinScoring'. The scripts for fluorescence data analysis are located in https://github.com/ccgradi72/singlemolecule2. All scripts are also available in the Source Data file. Source data are provided with this paper.

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

## Acknowledgements

We thank and acknowledge Mickael Krzeminski and Rhea Hudson for useful discussions during the execution of this project and Lewis Kay and Ranjith Muhandiram for their expert help with NMR spectroscopy. All ITC and DSC experiments were performed at The Hospital for Sick Children's Structural & Biophysical Core Facility with the assistance of Greg Wasney. The phosphorylation status of each 4E-BP2 mutant was confirmed at the AIMS Mass Spectrometry Laboratory, Department of Chemistry, University of Toronto. This work was funded by the Canadian Institutes of Health Research (CIHR, MOP-119579, to J.D.F.-K.) and the Natural Sciences and Engineering Research Council of Canada (NSERC, RGPIN 2017—06030 to C.G). A.B. was partly supported by a CIHR post-doctoral fellowship, and Z.Z. was supported by a doctoral CIHR Training Grant.

## Author contributions

J.E.D., A.B., and J.D.F.-K. conceived the project and designed and analyzed the NMR, ITC, and DSC experiments, with contributions from N.S. C.G., and Z.Z. designed and analyzed the smFRET and FCS experiments. J.E.D., A.B., H.L., and M.V. prepared reagents. J.E.D., A.B., Z.Z. and P.A.C. performed experiments. R.M.V. conducted the bioinformatic analysis. J.E.D. wrote the manuscript with contributions from A.B., R.M.V. and Z.Z. J.E.D., A.B., R.M.V., N.S., C.G. and J.D.F.-K. edited the manuscript.

## Competing interests

The authors declare no competing interests.
