## [Peer Review File · Nature Communications]

Reviewers' comments:

Reviewer #1 (Remarks to the Author):

Dawson et al have performed a thorough structural analysis of the effect of phosphorylation of eIF4E-binding proteins (4E-BPs) on their interaction with eIF4E. In particular, they demonstrate that the binding affinity of 4E-BPs for eIF4E is modulated by the stabilization of a 4E-BP beta-fold domain, which sequesters the eIF4E binding site, by hierarchical phosphorylation at five sites. The folding is non-cooperative, resulting in a tuneable equilibrium between the folded state and a non-folded, alpha-helical state. By fully characterizing the effects of each of the five phosphosites, the authors could reject a previously proposed model of phospho-inhibition of 4E-BP:eIF4E binding. They further demonstrate by bioinformatics methods that two hairpin structures in the beta fold are representative of similar structures found in other proteins and can be directly transplanted into them to provide phospho-inducible control of protein stability.

Overall I find the work to be important for settling an unresolved question, and I believe the overall conclusions to be sound. It also provides an interesting model case of functional impact of hierarchical phosphorylation, notably in an intrinsically disordered protein, which is of general interest in protein signaling and regulation. The identification of a transplantable, phospho-regulatory motif for protein stabilization and degradation would also be interesting for synthetic biology. The text overall is clear and easy to understand, despite protein structural analysis being outside the scope of my expertise.

However, I have a few concerns, particularly about the bioinformatics methods used in attempting to broaden the scope to other proteins. I will not assess the protein structure analyses aside from a couple minor comments.

1. The description of the bioinformatics methods is frustratingly vague and confusingly worded

Unfortunately, the methods as written are such that I am not 100% sure that I know what the authors did and I would not know how to replicate it. This makes it difficult to assess the methods' appropriateness. The rest of the supplementary methods employ clear mathematical formulae and occasional textual examples or reasoning, and thus are easy to follow. I think the bioinformatics methods should be rewritten in the same manner.

Unfortunately, as far as I can tell, the code for this analysis is not available alongside the other code, which would have helped in interpreting the written methods.

For example:

"To control for redundant structures in our dataset we then normalized motif observations by calculating the frequency of observation for every unique 11 residue window found in the PDB...This results in 8793 unique 11 residue motifs found in hairpin configuration, each scored by the number of hairpin observations divided by the total number of matching motif observations in the PDB."

What does this mean? Is the normalization in the first sentence distinct from the operation described in the second sentence or are these one in the same? Am I correct in interpreting "frequency of observation for every unique 11 residue window" to mean "position-specific (i-3 to i+7) amino acid frequencies for all 11-residue windows from the entire proteome"? What precisely constitutes a "hairpin observation" and a "matching motif observation"? Because it is not clear what operation is actually being carried out, I do not know how it is proposed to normalize the results in order to control for redundant structures.

In the following sentence, "...we took the sum of frequency values...", does "frequency values" refer to the ratio calculated in the previous paragraph?

"The hairpin score for a given xxxTPxGxxx or xxxSPxGxxx sequence is then defined as...": according to Figure 8A, the authors also observed some aspartic acid, asparagine, cysteine and methionine residues in the +3 position. Do these sequences not have hairpin scores then?

In any case, after some deciphering, I gather that the authors are calculating an empirical position-specific frequency value for each amino acid in hairpin structures. However, I am then completely flummoxed by the statement that the scores are defined "by averaging the frequency values of the amino acids at each x position over motifs that match either TPxG or SPxG". I can think of too many interpretations. Is it simply the mean hairpin-residue frequency for each residue observed in a given sequence, not counting positions i, i+1 and i+3? If so, that could be made much clearer through simple mathematical notation.

One particular comment is that the word "motif" appears to have multiple meanings across the three short paragraphs. I suggest reserving it for one precise case and finding different words for the other cases. For example, for "motifs found in Phosphosite database", I think the authors are

referring to what might more commonly be referred to as "sequence windows" (a single sequence is not a motif). Would that not be more accurately worded as "by scoring the sequence windows found in the PhosphoSitePlus database that match the xxxTPxGxxx and xxxSPxGxxx motifs, for both the human and mouse proteomes"?

2. Common methods already exist for such bioinformatics scoring

If the above interpretation of the hairpin score (mean frequency value) is correct, I do not think it has a solid statistical foundation: each frequency value is drawn from a different distribution depending on the position and there is no reason to believe that the results should be normally distributed. Plus, some positions are more well-resolved (i.e. more extreme frequency values) than others, resulting in a sort of position-based weighting effect. It would make more sense to treat the frequencies as empirical probabilities and to formulate the score in terms of probability/likelihood of observing a given sequence in a hairpin structure.

Well-established methods already exist to score a given sequence against a motif in this way. Such position-specific scoring matrices (PSSMs, aka position weight matrices or PWMs) have been around for decades and have been particularly common since the introduction of PSI-BLAST.

In this manner, the authors' score should be reformulated either as the product of the position-specific probabilities of observing each residue in a sequence or (more commonly) as the sum of their log-likelihoods (measured against some background/expected probabilities). There are countless suitable references, but consider, e.g. Altschul et al. 1997. *Nucleic Acids Res.* 25(17): 3389-3402 for an overview or Yaffe et al. 2001. *Nature Biotech.* 19: 348-353 for an application of the method to phosphosite prediction.

Sequence-weighting methods also exist to reduce the influence of redundant sequences when calculating empirical residue frequencies. See, e.g., Henikoff & Henikoff. 1994. *J. Mol. Biol.* 243: 574–578, which is used in PSI-BLAST and is easy to implement.

Finally, the information contained in Figure 8A would more commonly be represented as a sequence logo (Schneider & Stephens. 1990. *Nuc. Acids Res.* 18 (20): 6097-6100), which is easily constructed via, e.g. WebLogo (Crooks et al. 2004. *Genome Res.* 14(6):1188-90) or ggseqlogo (Wagih. 2017. 33(22):3645-7). This would be more informative and quicker to interpret than a frequency heatmap.

Using such existing, well-supported methods would be preferable to what appear to be ad hoc methods. Otherwise the authors should explain more clearly the reasoning behind their approach and why the established, common methods are unsuitable.

3. The hairpin score performance should be cross-validated

The authors should assess how well the hairpin scoring method identifies known hairpin structures before applying it to new sequences. Otherwise, we cannot know how many false positives to expect to find in their results. That, in turn, would provide a way of assessing what a reasonable score threshold would be. This could be achieved through a ROC (Receiver Operating Characteristic; false positive rate vs true positive rate) analysis and a precision-recall analysis. It should be done under a suitable cross-validation regime, e.g. k-fold cross validation, against some negative cases (such as 11-residue sequence windows matching the [T/S]PxG motif that are known not to be a part of a hairpin structure).

4. Minor concerns

- The authors refer to the "Phosphosite database" (in Results, Supplemental Methods and Figure 8 legend), which should more accurately be called the "PhosphoSitePlus database". Also, in the supplemental methods, the citation for the database is missing.

- I think Figure 6 could be made a lot easier to read and interpret if the "A-M" variant labels along the x-axis could be replaced with the protein diagrams from Supp. Figure 6A.

- The Figure 6 legend should note what the uncertainty bars represent.

- Why are different species sets used to illustrate sequence conservation across the different 4E-BPs in Supp. Figure 1A? I would expect the same species to be used for each comparison, unless a given isoform is not present in some species' genome.

- The authors state that "We find ... phospho-serine is also capable of forming a hairpin, both in the independent SPGGT motif introduced in the C-IDR as well as in a ... double mutant". Given this statement, the mutational accessibility of serine from threonine, and the alignments in Supp Figure 1A, I was left wondering why the threonines at 37/46 are so conserved and serines are not also

observed in those positions. The answer is buried in the Supplemental Figure 7 legend, where they state that the Gly peaks are "not as down-field shifted" in the double mutant as in the WT, "suggesting that the latter is more stable". I think that it is important to make this point about stability in the main text.

- The heatmap in Figure 8A uses a diverging color scheme centered on 0.5. A linear scheme (e.g. 0.0 = dark to 1.0 = light) would be more natural unless 0.5 carries a special meaning.

- For Figures 8C & D, the size of the bootstrap sample (i.e. how many proteins sampled per decile per iteration) should be reported.

Reviewer #2 (Remarks to the Author):

The authors present an impressive set of biophysical data from an array of different techniques to convincingly illustrate a graded regulatory mechanism by which hierarchical phosphorylation modulates domain folding stability and conformational ensemble in the 4E-BP2:eIF4E system. This work is a nice continuation from their earlier work, and clearly showcases how phosphorylation in the C-terminal intrinsically disordered region (IDR) in 4E-BP2 fine-tunes the accessibility of a canonical eIF4E-binding motif, therefore affecting the binding affinity of eIF4E. To systematically study the effect of phosphorylation, the authors create a number of phosphorylation constructs and use a phosphatase control to rule out a remote possibility that the changes in chemical shifts are due to Ala mutations. Remarkably, the authors further show that the same hairpin turn (pTPGGT) can be transplanted into other proteins as a phosphorylation-regulation unit, demonstrating the generalizability of their finding. Overall, this manuscript is well written, and the finding is important and significant. Therefore, in my opinion, this manuscript is suitable for Nature Communication. I would like the authors to consider the following minor points to further polish their work:

1. The authors use several denaturation conditions to study the non-cooperative stability of phosphorylated 4E-BP2 by NMR. I notice that a pH titration has been done for the non-phosphorylated 4E-BP2 in their previous paper (ref 17). However, to enable a direct comparison, I suggest the authors perform at least one identical denaturation experiment to the non-phosphorylated 4E-BP2 by NMR. Alternatively, they can measure DSC for the non-phosphorylated 4E-BP2, which could match to Figure 7C for FAF1.

2. The authors calculate the Rosetta minimization energies for the 4E-BP2:eIF4E structures with and without phosphorylation, and conclude that the electrostatic differences cannot explain the binding behavior. Due to the inherent dynamic nature of the binding process, this conclusion solely based on

computation is not quite convincing. I suggest the authors corroborate this by measuring the K_d of 4E-BP2:eIF4E interaction as a function of salt concentration.

3. The authors should clarify how the error was calculated for experimental observables throughout the manuscript. For example, the errors in the apparent pK_a , K_d , R_H , and FRET $\langle E \rangle$ efficiency. Are they based on the bootstrap errors calculated for the bioinformatic estimates in Figure 8 or experimental errors?

4. Maybe it is helpful to label both the N and C ends as well as G39 and G48 in the structure in Figure 2 for a better readability.

5. The authors should define the “apparent pK_a ” plotted in Figure 2A for residues that are not two-state folders (i.e., residues fitted with more than one pK_a).

6. The spectra in Figure 7A and D are too small to see clearly.

7. In order to derive Eq. S12, the orientation factor κ is assumed to average out during smFRET and necessary control experiments will be needed to justify it. Since Eq. S12 was not used in any figures, I suggest removing Eq. S12.

Reviewer #3 (Remarks to the Author):

Dawson et al describe studies of non-cooperative folding of different phosphorylation isoforms of eIF4E binding protein (4E-BP). They find that phosphorylation induces partial folding of 4E-BPs that hinders 4E-BPs binding to eIF4E. Although the authors use an impressive array of assays, from my point of view, current manuscript adds little to the authors’ previous publication (Bah, A. et al. Nature 2015) and is more suitable for publication in more specialized journal than Nat Comm.

In addition, the manuscript is written in cumbersome and intricate way. It contains multiple examples of convoluted sentences that are hard to follow (e.g. P. 11 “This indicates that the average distance between residue 32 in the folded domain region and residue 91 in the C-IDR increases upon two-phospho-induced folding and again upon stabilization of folding, consistent with helical conformations sampled in non-phospho 4E-BP2 bringing dyes at residues 32 and 91 into close average proximities, and increasing phosphorylation shifting the conformational equilibrium towards more extended, β -strand-like conformations that pull the dyes farther apart”).

Some interpretations of experimental data are also convoluted and seem questionable. For example, shifts in FRET toward lower values observed in the presence of GdmCl are interpreted as evidence of

protein unfolding while shifts in FRET toward lower values observed in the absence of denaturants are interpreted as evidence of folding. If 4E-BP folding and unfolding produce similar changes in distribution of FRET values doesn't it make this FRET assay uninformative and uninterpretable?

Responses to Reviewer comments

We appreciate all the reviewers' comments and suggestions, which have significantly improved the manuscript. In what follows, the reviewers' comments are given in *italics* (Arial, 11 point) with our response in regular text (Arial, 12 point).

Reviewer #1 (Remarks to the Author):

Dawson et al have performed a thorough structural analysis of the effect of phosphorylation of eIF4E-binding proteins (4E-BPs) on their interaction with eIF4E. In particular, they demonstrate that the binding affinity of 4E-BPs for eIF4E is modulated by the stabilization of a 4E-BP beta-fold domain, which sequesters the eIF4E binding site, by hierarchical phosphorylation at five sites. The folding is non-cooperative, resulting in a tuneable equilibrium between the folded state and a non-folded, alpha-helical state. By fully characterizing the effects of each of the five phosphosites, the authors could reject a previously proposed model of phospho-inhibition of 4E-BP:eIF4E binding. They further demonstrate by bioinformatics methods that two hairpin structures in the beta fold are representative of similar structures found in other proteins and can be directly transplanted into them to provide phospho-inducible control of protein stability.

Overall I find the work to be important for settling an unresolved question, and I believe the overall conclusions to be sound. It also provides an interesting model case of functional impact of hierarchical phosphorylation, notably in an intrinsically disordered protein, which is of general interest in protein signaling and regulation. The identification of a transplantable, phospho-regulatory motif for protein stabilization and degradation would also be interesting for synthetic biology. The text overall is clear and easy to understand, despite protein structural analysis being outside the scope of my expertise.

However, I have a few concerns, particularly about the bioinformatics methods used in attempting to broaden the scope to other proteins. I will not assess the protein structure analyses aside from a couple minor comments.

1. The description of the bioinformatics methods is frustratingly vague and confusingly worded

Unfortunately, the methods as written are such that I am not 100% sure that I know what the authors did and I would not know how to replicate it. This makes it difficult to assess the methods' appropriateness. The rest of the supplementary methods employ clear mathematical formulae and occasional textual examples or reasoning, and thus are easy to follow. I think the bioinformatics methods should be rewritten in the same manner.

Unfortunately, as far as I can tell, the code for this analysis is not available alongside the other code, which would have helped in interpreting the written methods.

For example:

"To control for redundant structures in our dataset we then normalized motif observations by calculating the frequency of observation for every unique 11 residue window found in the PDB... This results in 8793 unique 11 residue motifs found in hairpin configuration, each scored by the number of hairpin observations divided by the total number of matching motif observations in the PDB."

What does this mean? Is the normalization in the first sentence distinct from the operation described in the second sentence or are these one in the same? Am I correct in interpreting "frequency of

observation for every unique 11 residue window" to mean "position-specific (i-3 to i+7) amino acid frequencies for all 11-residue windows from the entire proteome"? What precisely constitutes a "hairpin observation" and a "matching motif observation"? Because it is not clear what operation is actually being carried out, I do not know how it is proposed to normalize the results in order to control for redundant structures.

In the following sentence, "...we took the sum of frequency values...", does "frequency values" refer to the ratio calculated in the previous paragraph?

"The hairpin score for a given xxxTPxGxxx or xxxSPxGxxx sequence is then defined as...": according to Figure 8A, the authors also observed some aspartic acid, asparagine, cysteine and methionine residues in the +3 position. Do these sequences not have hairpin scores then?

In any case, after some deciphering, I gather that the authors are calculating an empirical position-specific frequency value for each amino acid in hairpin structures. However, I am then completely flummoxed by the statement that the scores are defined "by averaging the frequency values of the amino acids at each x position over motifs that match either TPxG or SPxG". I can think of too many interpretations. Is it simply the mean hairpin-residue frequency for each residue observed in a given sequence, not counting positions i, i+1 and i+3? If so, that could be made much clearer through simple mathematical notation.

"One particular comment is that the word "motif" appears to have multiple meanings across the three short paragraphs. I suggest reserving it for one precise case and finding different words for the other cases. For example, for "motifs found in Phosphosite database", I think the authors are referring to what might more commonly be referred to as "sequence windows" (a single sequence is not a motif). Would that not be more accurately worded as "by scoring the sequence windows found in the PhosphoSitePlus database that match the xxxTPxGxxx and xxxSPxGxxx motifs, for both the human and mouse proteomes"?"

We have taken the criticism in points 1 and 2 as an opportunity to completely redo the bioinformatics section. We now use a standard Position-Specific Scoring Matrix (PSSM) for scoring 11 residue sequence windows, and have fully replaced the methods section and Figure 8 based on a ground up rework of the analysis. We note that while specific details have changed, the overall results and conclusions remain broadly the same, with the PSSM based method agreeing in a robust way with the old, now-unused method (which was based partially on sequence and partially on the concept of "structural occupancy" for sequences solved multiple times in multiple contexts). Specifically, enrichment of ubiquitination-related features for high hairpin propensity sequences is still observed, and the rationale for selecting FAF1 as a TPGGT transplant candidate was maintained. The standard method suggested by Reviewer #1 is easier to explain, easier to understand, and has a clear statistical foundation.

Importantly, a repository of code and data used for the new PSSM based method is now provided as a supplemental file.

The new PSSM based method scores every position and is trained on all hairpin containing sequences (without restricting any position). However, for measuring enrichment of ubiquitination related markers, we only score sequences matching the xxx[T/S]PxGxxxx filter, because we are specifically testing 4E-BP like proline directed phosphosites. While aspartic acid, asparagine, cysteine, and methionine are present at the +3 position at low frequency in folded structures, we don't observe hairpins in 4E-BP2 without glycine at that position, so the basis of their formation is not 4E-BP2 like.

The TPGGT motif itself is stable enough to form hairpins at some occupancy even in the context of an otherwise entirely disordered protein, but other sequences, even sequences that are too unstable to form hairpins on their own, could be forced into the same

conformation by the cooperative folding of an ordered protein domain. The behavior of these non-4E-BP2 like hairpins is an interesting question, but we feel it is outside the scope of this study.

The text was also edited throughout to use the words “motif” and “sequence” more precisely.

2. Common methods already exist for such bioinformatics scoring

If the above interpretation of the hairpin score (mean frequency value) is correct, I do not think it has a solid statistical foundation: each frequency value is drawn from a different distribution depending on the position and there is no reason to believe that the results should be normally distributed. Plus, some positions are more well-resolved (i.e. more extreme frequency values) than others, resulting in a sort of position-based weighting effect. It would make more sense to treat the frequencies as empirical probabilities and to formulate the score in terms of probability/likelihood of observing a given sequence in a hairpin structure.

Well-established methods already exist to score a given sequence against a motif in this way. Such position-specific scoring matrices (PSSMs, aka position weight matrices or PWMs) have been around for decades and have been particularly common since the introduction of PSI-BLAST.”

*In this manner, the authors' score should be reformulated either as the product of the position-specific probabilities of observing each residue in a sequence or (more commonly) as the sum of their log-likelihoods (measured against some background/expected probabilities). There are countless suitable references, but consider, e.g. Altschul et al. 1997. *Nucleic Acids Res.* 25(17): 3389-3402 for an overview or Yaffe et al. 2001. *Nature Biotech.* 19: 348-353 for an application of the method to phosphosite prediction.*

*Sequence-weighting methods also exist to reduce the influence of redundant sequences when calculating empirical residue frequencies. See, e.g., Henikoff & Henikoff. 1994. *J. Mol. Biol.* 243: 574–578, which is used in PSI-BLAST and is easy to implement.*

*Finally, the information contained in Figure 8A would more commonly be represented as a sequence logo (Schneider & Stephens. 1990. *Nuc. Acids Res.* 18 (20): 6097-6100), which is easily constructed via, e.g. WebLogo (Crooks et al. 2004. *Genome Res.* 14(6):1188-90) or ggseqlogo (Wagih. 2017. 33(22):3645-7). This would be more informative and quicker to interpret than a frequency heatmap.*

Using such existing, well-supported methods would be preferable to what appear to be ad hoc methods. Otherwise the authors should explain more clearly the reasoning behind their approach and why the established, common methods are unsuitable.

We appreciate this criticism and have recalculated the score using a PSSM, as described above for point 1. We have changed the score to use a sum of log-likelihoods, measured against the background probability (position-independent) of the hairpin-containing 11mers themselves.

We have added a frequency sequence logo as Figure 8B. We have chosen not to use the information content as the y-axis because the value at position 3 (the glycine in [T/S]PxG) is so much larger than the others that it becomes the only legible letter in the logo.

3. The hairpin score performance should be cross-validated

The authors should assess how well the hairpin scoring method identifies known hairpin structures before applying it to new sequences. Otherwise, we cannot know how many false positives to expect

to find in their results. That, in turn, would provide a way of assessing what a reasonable score threshold would be. This could be achieved through a ROC (Receiver Operating Characteristic; false positive rate vs true positive rate) analysis and a precision-recall analysis. It should be done under a suitable cross-validation regime, e.g. k-fold cross validation, against some negative cases (such as 11-residue sequence windows matching the [T/S]PxG motif that are known not to be a part of a hairpin structure).

ROC analysis of 5-fold cross validation has been added as Figure 8C. For PDB derived 11-residue sequence windows matching the [T/S]PxG motif, we observe a roughly 1:4 ratio of hairpin structures to non-hairpins structures. The true/false positive rates were tested with 100 repeats of 5 fold cross validation (500 curves in total) with training/testing data split based on a list of 3870 nonredundant PDB chains. During the 5-fold cross validation, the PSSM method averages an area under the curve 0.82 for discriminating hairpins from non-hairpins, demonstrating that the method is generally predictive and reasonable for the task of testing enrichment.

4. Minor concerns

- The authors refer to the "Phosphosite database" (in Results, Supplemental Methods and Figure 8 legend), which should more accurately be called the "PhosphoSitePlus database". Also, in the supplemental methods, the citation for the database is missing.

The title of database has been corrected in all three locations, and the reference is added to the supplemental methods.

- I think Figure 6 could be made a lot easier to read and interpret if the "A-M" variant labels along the x-axis could be replaced with the protein diagrams from Supp. Figure 6A."

We thank Reviewer #1 for this very useful idea. We added the phospho-site diagrams, oriented vertically, to the bottom of Figure 6. We also removed the three schematics at the top.

- The Figure 6 legend should note what the uncertainty bars represent.

New text has been added in the Figure 6 legend, including information about new schematics: "The K_D values and their uncertainties are from this work or Bah et al (Nature, 2015) (see Suppl. Figure 6A) and were measured using ITC (see Supplemental Information) as the mean and standard deviation of sample repeats. Individual K_D values found for this paper are indicated with orange diamonds. The affinities for non-, two-, and five-phospho 4E-BP2 are marked in the chart with blue circles. Schematic cartoons for each of the 4E-BP2 variants are shown at the bottom of the chart in which green circles indicate non-phosphorylated sites, purple circles are phosphorylated residues, and the tan oval represents the folded domain of 4E-BP2."

New text has been added in Supplemental Information: "The K_D values and their uncertainties are from this work or Bah et al² (see Suppl. Figure 7A). Repeat data collections (n=3 for five-phospho 4E-BP2; n=2 for pT37/pT46/p**T70**, pT37/pT46/p**S65**, pT37/pT46/p**T70**/p**S83**, pT37/pT46/p**S65**/p**S83**, pT37/pT46/p**S65**/p**T70**; n=1 for pT37/pT46/p**S83** due to challenges in sample preparation and stability, as well as data quality) were taken from different samples, some from the same preparation and others from different preparations. The K_D values and their uncertainties were estimated as the mean and standard deviations. The WT 5-phospho 4E-BP2 K_D value calculated for this article is comparable to that from Bah et al."

- Why are different species sets used to illustrate sequence conservation across the different 4E-BPs

in Supp. Figure 1A? I would expect the same species to be used for each comparison, unless a given isoform is not present in some species' genome.

The 4E-BP1, 4E-BP2, and 4E-BP3 entries were replaced with entries from a consistent set of seven species: human, mouse, rat, beaver, pig, cat, and dog. The UniProtKB identifiers are also included with the label. We have no means to ascertain that the 4E-BP1 example from the black Snub-nose monkey genome is not a mis-sequencing and therefore removed it. Replacing it are examples of invertebrate 4E-BP proteins, which have only one known isoform (Kume *et al*, 2012, *Parasitol Res*; Lasko P, 2000, *J. Cell Biol.*). The invertebrates lack the S83 phospho site and ticks have a TPGGS sequence instead of a TPGGT sequence at the second hairpin site. The text has been updated to include this new information, as well as referencing to the Uniprot database and Clustal Omega sequence alignment program.

- The authors state that "We find ... phospho-serine is also capable of forming a hairpin, both in the independent SPGGT motif introduced in the C-IDR as well as in a ... double mutant". Given this statement, the mutational accessibility of serine from threonine, and the alignments in Supp Figure 1A, I was left wondering why the threonines at 37/46 are so conserved and serines are not also observed in those positions. The answer is buried in the Supplemental Figure 7 legend, where they state that the Gly peaks are "not as down-field shifted" in the double mutant as in the WT, "suggesting that the latter is more stable". I think that it is important to make this point about stability in the main text. We agree and have now added this to the main text: "However, the characteristic ^{39/48}Gly peaks are not as down-field shifted in the double mutant as in the WT, suggesting that the latter is more stable".

- The heatmap in Figure 8A uses a diverging color scheme centered on 0.5. A linear scheme (e.g. 0.0 = dark to 1.0 = light) would be more natural unless 0.5 carries a special meaning. The heatmap now ranges from 0.0 (dark) to 1.0 (light).

- For Figures 8C & D, the size of the bootstrap sample (i.e. how many proteins sampled per decile per iteration) should be reported.

The following passage has been added to the Supplementary Information: "Enrichment was measured by comparing $\log_2(O/E)$ for the frequency of observed features in high scoring sets of sequences divided by the frequency over the full set. Significance of enrichment was tested using bootstrap analysis, sampling the 6619 human and 5084 mouse proteins (with replacement) 10000 times to calculate the 99% confidence intervals. Doing this across multiple PSSM score thresholds shows that there is significant enrichment ($p < 0.01$) for the top 20% by PSSM score over all three properties for both human and mouse (Figure 8D-I), and for 5 out of 6 of the top 10%."

Reviewer #2 (Remarks to the Author):

The authors present an impressive set of biophysical data from an array of different techniques to convincingly illustrate a graded regulatory mechanism by which hierarchical phosphorylation modulates domain folding stability and conformational ensemble in the 4E-BP2:eIF4E system. This work is a nice continuation from their earlier work, and clearly showcases how phosphorylation in the C-terminal intrinsically disordered region (IDR) in 4E-BP2 fine-tunes the accessibility of a canonical eIF4E-binding motif, therefore affecting the binding affinity of eIF4E. To systematically study the effect of phosphorylation, the authors create a number of phosphorylation constructs and use a phosphatase control to rule out a remote possibility that the changes in chemical shifts are due to Ala mutations. Remarkably, the authors further show that the same hairpin turn (pTPGGT) can be transplanted into other proteins as a phosphorylation-regulation unit, demonstrating the generalizability of

their finding. Overall, this manuscript is well written, and the finding is important and significant. Therefore, in my opinion, this manuscript is suitable for Nature Communication. I would like the authors to consider the following minor points to further polish their work:

1. The authors use several denaturation conditions to study the non-cooperative stability of phosphorylated 4E-BP2 by NMR. I notice that a pH titration has been done for the non-phosphorylated 4E-BP2 in their previous paper (ref 17). However, to enable a direct comparison, I suggest the authors perform at least one identical denaturation experiment to the non-phosphorylated 4E-BP2 by NMR. Alternatively, they can measure DSC for the non-phosphorylated 4E-BP2, which could match to Figure 7C for FAF1.

We agree with the reviewer that at least one identical denaturation experiment of the non-phospho 4E-BP2 is needed to do a direct comparison with phospho 4E-BP2. Therefore, we have collected a set of HSQC spectra for non-phospho 4E-BP2 at pH 5.0 between 5 and 70°C to match one of our thermal denaturation of five-phospho 4E-BP2. The behavior of the disordered protein's peaks under thermal denaturation (the expected upfield reference shift) was used in the text to contrast the shifts towards random coil chemical shifts ($\omega_{1H} \sim 8.2\text{ppm}$) observed during denaturation.

2. The authors calculate the Rosetta minimization energies for the 4E-BP2:eIF4E structures with and without phosphorylation, and conclude that the electrostatic differences cannot explain the binding behavior. Due to the inherent dynamic nature of the binding process, this conclusion solely based on computation is not quite convincing. I suggest the authors corroborate this by measuring the K_D of 4E-BP2:eIF4E interaction as a function of salt concentration.

The authors apologise for the confusion. There is still a counter-hypothesis in the field, based on X-ray structures of eIF4E binding to 4E-BP peptides, that the difference in K_D between non-phosphorylated and phosphorylated 4E-BPs is due to electrostatic repulsion between pS65 and the negative charge on the eIF4E surface. The point we were making was that this conclusion could not be made based on the crystal structures. The rest of the manuscript, as well as the ITC data from the previous paper, demonstrates that phosphorylation acts to increasingly lock the eIF4E-binding site away in a binding-incompetent form. The mechanism is based on changing conformational propensities, not electrostatics. We have clarified the text throughout the manuscript and believe that this confusion is now resolved.

3. The authors should clarify how the error was calculated for experimental observables throughout the manuscript. For example, the errors in the apparent pK_a , K_D , R_H , and FRET $\langle E \rangle$ efficiency. Are they based on the bootstrap errors calculated for the bioinformatic estimates in Figure 8 or experimental errors?

We added and clarified error analysis information throughout, including in the main text, supplementary information, and figure legends, some of which is listed in the response to Reviewer #1.

4. Maybe it is helpful to label both the N and C ends as well as G39 and G48 in the structure in Figure 2 for a better readability.

We have added the additional information, which does improve readability.

5. The authors should define the “apparent pK_a ” plotted in Figure 2A for residues that are not two-state folders (i.e., residues fitted with more than one pK_a).

The apparent pK_a values characterize two state denaturation events in which protonation and denaturation occur simultaneously. We added information to the main text on how we distinguished between titrations involved in denaturation and those that were not.

6. The spectra in Figure 7A and D are too small to see clearly.

We have changed the layout of Figure 7 to make the spectra larger.

7. In order to derive Eq. S12, the orientation factor $Kappa$ is assumed to average out during smFRET and necessary control experiments will be needed to justify it. Since Eq. S12 was not used in any figures, I suggest removing Eq. S12.

With respect to the Kappa-squared factor, we have done control measurements (Zhang, 2017, Single-Molecule Spectroscopy of Disordered States and Dynamics in Proteins) and found out that the half-cone angle describing the rotational freedom of the dye is fairly large (~75-80 degrees) at all six labelling positions, including the two used for FRET. As per our theoretical study (Badali and Gradinaru, 2011, J. Chem. Phys.), this ensures that the 2/3 approximation for kappa-squared factor in the Forster radius is valid. Having said that, we do not mention inferred interdye distances explicitly in this manuscript, so Reviewer #2's suggestion of removing Eq. S12 was followed. We do, however, mention that the dyes retained considerable rotational freedom under all conditions investigated and therefore the reported FRET E values are good measures of proximity distances and can be directly compared among various conditions tested.

Reviewer #3 (Remarks to the Author):

Dawson et al describe studies of non-cooperative folding of different phosphorylation isoforms of eIF4E binding protein (4E-BP). They find that phosphorylation induces partial folding of 4E-BPs that hinders 4E-BPs binding to eIF4E. Although the authors use an impressive array of assays, from my point of view, current manuscript adds little to the authors' previous publication (Bah, A. et al. Nature 2015) and is more suitable for publication in more specialized journal than Nat Comm.

We would like to acknowledge Review #3 for this comment, which highlights significant confusion that we have now addressed. The authors respectfully disagree that this current manuscript adds little to our previous publication (Bah et. al. Nature 2015). In that paper, we described the discovery of the phosphorylation induced folding of 4E-BP2, which inhibits binding to eIF4E. However, the mechanism by which this regulation happens or whether this mechanism is used by other proteins was not known. The data in the current paper are highly novel, as we used an array of assays to demonstrate the mechanism behind the three-stage attenuation of eIF4E binding that is at the core of the phospho-regulation of 4E-BP2 inhibition and, very importantly, why the effects of C-IDR phosphorylation build on the presence of the folded domain. We showed that the folding of the phosphorylation-induced fold is non-cooperative, resulting in a tunable equilibrium between the folded phosphorylated beta-state and the unfolded non-phosphorylated alpha state. We now have made this point much clearer in the text and included a diagram outlining these points (Figure 6B). Furthermore, our bioinformatics data demonstrate that the two hairpins in phospho-4EBP2 are similar to hairpins found in other proteins. Finally, we demonstrate that we can transplant the phospho hairpins, thereby generating a phospho-inducible control of protein stability. Therefore, this paper should be of significant interest to the general readership of Nature Com. in the fields of structural biology, synthetic biology and PTM-mediated signaling and regulation.

In addition, the manuscript is written in cumbersome and intricate way. It contains multiple examples of convoluted sentences that are hard to follow (e.g. P. 11 "This indicates that the average distance between residue 32 in the folded domain region and residue 91 in the C-IDR increases upon two-phospho-induced folding and again upon stabilization of folding, consistent with helical conformations

sampled in non-phospho 4E-BP2 bringing dyes at residues 32 and 91 into close average proximities, and increasing phosphorylation shifting the conformational equilibrium towards more extended, β -strand-like conformations that pull the dyes farther apart”).

We thank the reviewer for this comment, and have rewritten this and other long and convoluted sentences found throughout the manuscript, making simpler and less intricate sentences. These changes have made the manuscript easier to read and follow.

Some interpretations of experimental data are also convoluted and seem questionable. For example, shifts in FRET toward lower values observed in the presence of GdmCl are interpreted as evidence of protein unfolding while shifts in FRET toward lower values observed in the absence of denaturants are interpreted as evidence of folding. If 4E-BP folding and unfolding produce similar changes in distribution of FRET values doesn't it make this FRET assay uninformative and uninterpretable?

We appreciate this comment and have now clarified the explanation of the smFRET data by discussing it more in the context of our complementary NMR data, which is measured on the same phosphorylation states and under identical conditions. Furthermore, we have performed additional Fluorescence Correlation Spectroscopy (FCS) experiments to rule out aggregation effects and to corroborate our FRET and NMR data. Finally, our new scheme (Figure 6B) depicting a cartoon representation of the non-, two- and five-phospho 4E-BP2 ensembles helps to further clarify results from our FRET data.

Briefly, it is an over-simplification to say that shifts to lower FRET (longer distance) can only mean one thing (i.e., denaturation) and that folding cannot produce the same result. This is what was observed in our smFRET experiments and corroborated by NMR measurements and represents what is unexpected and surprising about the system. For example, we found that the FRET efficiency increases as the pH decreases for all the three phospho states studied, but the underlying mechanism is quite different for each phospho state. In the non-phospho 4E-BP2, this is due to the stabilization of the transient helical conformations. In contrast, for the two- and for the five-phospho states, it is due to the destabilization of the extended conformations present in the beta folded structure and in the C-IDR as a result of the protonation of the phosphate groups. This destabilization of extended conformations is followed by the subsequent induction of transient helical conformations (Figure 4). As for the FRET data during the GdmCl denaturation, the non-, two- and five-phospho 4E-BP2 all follow a non-cooperative unfolding pathway for their ensemble of states as the GdmCl concentration is increased. Therefore, depending on the mechanism of denaturation (acid vs GdmCl) and the phosphorylation status of 4E-BP2, the final 'denatured' state of the protein could be very different. We could tease out this intricate mechanism because we have used an array of highly sensitive biophysical tools (including smFRET), which provide different but complementary information about this rather complex (but interesting) mechanism of phospho-regulation.

REVIEWERS' COMMENTS:

Reviewer #1 (Remarks to the Author):

The authors have satisfactorily addressed all of my previous concerns.

One minor comment. In the Supplemental Methods "Bioinformatic assessment of pTPGGT motif structure and sequences" section, the following sentence should be rewritten (or at least include an extra comma after "PSSM"), because its meaning was not immediately clear to me:

"Now, while the negative data does not overlap with the sequences used to define the PSSM the positive data does, having come from 570 of the 5611 nonredundant protein clusters identified previously."

This could be simplified to "Unlike the negative data, the positive data overlap with the sequences used to define the PSSM. This is because the positive data come from 570 of the ..."

Reviewer #2 (Remarks to the Author):

The new denaturation experiments of non-phosphorylated 4E-BP2 in Fig. S2B clearly show a uniform upfield shift with increasing temperature, in contrast to phosphorylated 4E-BP2. Therefore, my previous comment regarding this control has been addressed. In addition, the authors clarified the sources of errors throughout the text, which was also a concern from Review #1. They also improved the readability of the FRET/FCS sections, removed the redundant equation S12 and added the necessary FCS equations for $R_{\{H\}}$. I just have two minor comments:

1. The color codes of temperatures need to be added in Fig. S2B legend or caption.
2. The github links to analysis scripts are not available at the moment. They should be available after publication.

To sum up, the authors have satisfactorily addressed my comments in this version.

Reviewer #3 (Remarks to the Author):

The authors have greatly improved manuscript clarity. They've also made a number of compelling arguments supporting their conclusions and more clearly outlined novelty of their work. I think that the manuscript can be accepted for publication in its current form.

I only have a minor suggestion. I still think that smFRET experiments with chemical denaturation make interpretation of smFRET experiments done in the absence of denaturants more convoluted. I would argue that smFRET experiments with chemical denaturation are redundant because NMR and DSC experiments already demonstrate non-cooperative unfolding of 4E-BP2. Therefore, smFRET experiments with chemical denaturation could be removed for clarity.

Response to reviewer final comments

We appreciate the reviewers' comments and suggestions, which have significantly improved the manuscript. In what follows, the reviewers' comments are given in *italics (Arial, 11 point)* with our response in regular text (Arial, 12 point).

REVIEWERS' COMMENTS:

Reviewer #1 (Remarks to the Author):

The authors have satisfactorily addressed all of my previous concerns.

The authors thank Reviewer 1 for their helpful suggestions, especially on the bioinformatics section.

One minor comment. In the Supplemental Methods "Bioinformatic assessment of pTPGGT motif structure and sequences" section, the following sentence should be rewritten (or at least include an extra comma after "PSSM"), because its meaning was not immediately clear to me:

"Now, while the negative data does not overlap with the sequences used to define the PSSM the positive data does, having come from 570 of the 5611 nonredundant protein clusters identified previously."

This could be simplified to "Unlike the negative data, the positive data overlap with the sequences used to define the PSSM. This is because the positive data come from 570 of the ..."

Thank you for the suggested phrasing, which we have now incorporated as it is helpful in clarifying the sentence.

Reviewer #2 (Remarks to the Author):

The new denaturation experiments of non-phosphorylated 4E-BP2 in Fig. S2B clearly show a uniform upfield shift with increasing temperature, in contrast to phosphorylated 4E-BP2. Therefore, my previous comment regarding this control has been addressed. In addition, the authors clarified the sources of errors throughout the text, which was also a concern from Review #1. They also improved the readability of the FRET/FCS sections, removed the redundant equation S12 and added the necessary FCS equations for $R_{\{H\}}$. I just have two minor comments:

1. The color codes of temperatures need to be added in Fig. S2B legend or caption.

The codes were added to the figure.

2. The github links to analysis scripts are not available at the moment. They should be available after publication.

The github link was fixed.

To sum up, the authors have satisfactorily addressed my comments in this version.

The authors would like to thank Reviewer 2 for their suggestions, particularly for suggesting the non-phospho 4E-BP2 experiment, which demonstrated the difference in behavior between folded and IDP residues during denaturation.

Reviewer #3 (Remarks to the Author):

The authors have greatly improved manuscript clarity. They've also made a number of compelling arguments supporting their conclusions and more clearly outlined novelty of their work. I think that the manuscript can be accepted for publication in its current form.

Thank you to Reviewer 3 whose suggestions greatly helped improve the readability of the text.

I only have a minor suggestion. I still think that smFRET experiments with chemical denaturation make interpretation of smFRET experiments done in the absence of denaturants more convoluted. I would argue that smFRET experiments with chemical denaturation are redundant because NMR and DSC experiments already demonstrate non-cooperative unfolding of 4E-BP2. Therefore, smFRET experiments with chemical denaturation could be removed for clarity.

Based on Reviewer #3's original comments in the first review, we clarified that the shifts in smFRET $\langle E \rangle$ to lower values upon both GdmCl denaturation and phosphorylation in the absence of denaturation are due to different structural effects on the ensemble and reworded to enhance overall clarity. Given the utility of the denaturation data for understanding that the single observed smFRET peaks reflect ensemble conformational averages, making them valuable for interpretation of the data in the absence of denaturants, we feel that removing these data would weaken the paper. We also note that the smFRET denaturation data are in the supplementary figures and not main text figures.